# SELA: Tree-Search Enhanced LLM Agents for Automated Machine Learning

## Abstract

Automated Machine Learning (AutoML) approaches encompass traditional methods that optimize fixed pipelines for model selection and ensembling, as well as newer LLM-based frameworks that autonomously build pipelines. While LLM-based agents have shown promise in automating machine learning tasks, they often generate low-diversity and suboptimal code, even after multiple iterations. To overcome these limitations, we introduce Tree-Search Enhanced LLM Agents (**SELA**), an innovative agent-based system that leverages Monte Carlo Tree Search (MCTS) to optimize the AutoML process. By representing pipeline configurations as trees, our framework enables agents to conduct experiments intelligently and iteratively refine their strategies, facilitating a more effective exploration of the machine learning solution space. This novel approach allows SELA to discover optimal pathways based on experimental feedback, improving the overall quality of the solutions. In an extensive evaluation across 20 machine learning datasets, we compare the performance of traditional and agent-based AutoML methods, demonstrating that SELA achieves a win rate of 65% to 80% against each baseline across all datasets. These results underscore the significant potential of agent-based strategies in AutoML, offering a fresh perspective on tackling complex machine learning challenges. The code will be open-sourced upon publication.

## 1 Introduction

Automated Machine Learning (AutoML) is a rapidly evolving field that seeks to automate the process of designing reliable machine learning solutions with minimal human intervention. Traditional AutoML frameworks, such as Auto-WEKA (Thornton et al., 2013), Auto-Sklearn (Feurer et al., 2015; 2020), AutoGluon (Tang et al., 2024b), and H2O AutoML (LeDell & Poirier, 2020), rely on predefined search spaces and routines. These frameworks primarily focus on optimizing hyperparameters and model ensembling to find the best model configuration. However, this fixed and static approach often lacks the adaptability needed to handle diverse and dynamic data scenarios, resulting in suboptimal performance in more complex settings. Additionally, the traditional focus on model training leaves other crucial stages of the machine learning pipeline, such as data preprocessing and feature engineering, underexplored, thereby limiting the overall effectiveness of these systems.

Recently, large language model (LLM)-based agents have emerged as promising tools for automating machine learning tasks by leveraging natural language processing capabilities to generate code. These systems typically begin with a natural language prompt describing the dataset and the problem, after which an LLM generates an end-to-end solution. Early efforts, such as Zhang et al. (2024), experimented with prompting LLMs to generate machine learning solutions, while Hong et al. (2024) introduced agents equipped with Hierarchical Graph Modeling and Programmable Node Generation to address complex and dynamic workflows. Despite these advances, LLM-based solutions often fall short in generating diverse and highly optimized workflows, as their search process remains limited to a single pass or trial. Without iterative refinement or the ability to explore alternative strategies, these solutions frequently converge on suboptimal results, even when multiple attempts are allowed.

A critical shortcoming of both traditional AutoML and LLM-based frameworks lies in their inability to mimic the nuanced problem-solving approach of human experts. When approaching a machine

learning task, an expert does not simply execute a fixed pipeline. Instead, they explore various potential configurations, systematically conduct experiments, analyze results, and iteratively refine their understanding of each component's effectiveness. This iterative, feedback-driven process allows experts to explore diverse solutions and improve them incrementally until they arrive at the optimal configuration.

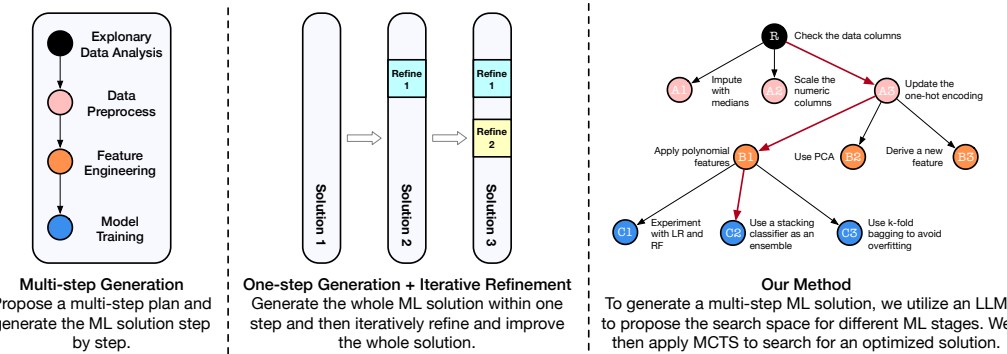

Figure 1: SELA's abstraction compared to other agent-based AutoML frameworks. There are two main types of agent-based approaches to AutoML problems. The first approach (Hong et al., 2024) divides a machine learning task into multiple stages, proposing a plan for each stage, and generating and executing code step by step according to the plan, with no refinement after the solution is completed. The second (Schmidt et al., 2024) generates the entire solution in one step and iteratively refines it as a whole. SELA integrates both approaches, enabling stage-wise planning while iteratively exploring better solutions at each stage level.

Inspired by this human-centered approach, we propose Tree-**S**earch **E**nhanced **L**LM **A**gents (**SELA**) for automated machine learning, a novel framework that integrates the strengths of LLM agents with a structured search and refinement process modeled on how experts solve machine learning problems. As illustrated in Figure 1, our framework combines the benefits of stage-wise planning, where each stage (e.g., Exploratory Data Analysis, Data Preprocessing, Feature Engineering, and Model Training) is handled sequentially, with an iterative refinement mechanism. In SELA, the search space of a machine learning problem is conceptualized as a tree, where each branch represents a potential solution path. This tree-based structure enables the agent to systematically explore and refine solutions, much like an expert who tests and improves their strategy based on continuous feedback.

To navigate this search space, we employ Monte Carlo Tree Search (MCTS) (Browne et al., 2012) as the core decision-making engine, leveraging its ability to balance exploration (testing new strategies) and exploitation (improving known good strategies). MCTS allows the agent to efficiently explore large decision spaces, collect and process experimental results, and intelligently select the next promising configuration to test. By iterating through this cycle of experimentation and refinement, SELA incrementally improves its solutions, offering a more dynamic and flexible approach than static AutoML frameworks.

We rigorously evaluated SELA using 20 diverse datasets from the AutoML benchmark, comparing its performance against both traditional AutoML systems and agent-based AutoML approaches. The results demonstrate that SELA consistently delivers superior performance across a wide range of machine learning tasks, validating its effectiveness and adaptability.

Our research makes the following contributions:

1. We introduce a novel approach that empowers LLM agents to address machine learning challenges through an iterative, feedback-driven process. This mirrors the methodology of human experts, enabling continuous exploration of various configurations and improving outcomes through multiple rounds of refinement. This iterative exploration yields more diverse and optimized solutions than single-pass strategies.

2. We present a robust system that intelligently selects and executes experiments to generate high-performance pipelines. At the heart of this framework is the conceptualization of the

machine learning search space as a tree, navigated using Monte Carlo Tree Search (MCTS). This approach allows the agent to systematically explore complex solution landscapes and adapt its strategy based on intermediate feedback, enabling the efficient discovery of effective solutions.

3. We provide a comprehensive comparison of agent-based AutoML systems with traditional AutoML frameworks, highlighting the significant untapped potential of agentic approaches in solving machine learning problems. Our findings suggest that this emerging paradigm offers a promising direction for future research, with considerable advantages in flexibility, adaptability, and performance.

## 2 RELATED WORKS

**Tree Search and Its Integration with LLMs** Tree search algorithms have significantly advanced problem-solving in artificial intelligence, with Monte Carlo Tree Search (MCTS) emerging as a leading technique. These algorithms have been successfully applied across various domains, including robotics (Best et al., 2019; Wu et al., 2015; Clary et al., 2018), chemistry (Segler et al., 2018), and gaming (Silver et al., 2016; 2017), where MCTS is used to navigate vast solution spaces and solve complex problems. More recently, research has focused on integrating tree search with Large Language Models (LLMs) to enhance reasoning and decision-making. Studies such as Krishnamurthy et al. (2024) and Dwaracherla et al. (2024) explored LLMs' capacities for efficient exploration, while Tang et al. (2024a) and Hui & Tu (2024) developed strategies for exploiting previously learned knowledge. Striking a balance between exploration and exploitation, Zhou et al. (2024) and Chi et al. (2024) applied MCTS for planning with external or self-evaluated feedback, while Feng et al. (2023); Wang et al. (2024) adapted AlphaZero-style tree search to LLM-based tasks. These advancements underscore the potential of combining tree search methods with LLMs, balancing exploration of new solutions with exploitation of prior knowledge to enhance decision-making.

**Advances and Limitations in AutoML Systems** Automated Machine Learning (AutoML) frameworks were introduced to reduce the need for expert knowledge in designing machine learning pipelines. Early AutoML efforts, such as (Feurer et al., 2020; Jin et al., 2019; Olson & Moore, 2016; Thornton et al., 2013), focused primarily on automating key pipeline components like hyperparameter optimization, model selection, and ensembling. These frameworks achieved notable progress by integrating meta-learning and hyperparameter search strategies to automatically select and tune machine learning models. More recent AutoML systems, such as (Erickson et al., 2020) and (LeDell & Poirier, 2020), employed ensembling techniques to further improve performance, and extensions into multi-modal data settings (Tang et al., 2024b; Jin et al., 2023) have broadened AutoML's applicability.

Recently, there has been growing interest in leveraging LLMs within AutoML systems to enhance pipeline flexibility. Studies such as Hollmann et al. (2024) and Li et al. (2024) applied LLMs to automate feature engineering, while Liu et al. (2024) introduced LLMs for hyperparameter tuning. In addition, Luo et al. (2024) proposed embedding LLMs at each stage of the machine learning workflow. Despite these advancements, traditional AutoML systems remain constrained by rigid pipelines and limited flexibility to adapt to unique datasets or specific task requirements.

**LLM Agents for Dynamic Machine Learning Pipelines** In contrast to static pipelines, LLM-based agents offer a more dynamic solution for addressing complex machine learning challenges. Hong et al. (2024) introduced an LLM agent with hierarchical graph modeling and programmable node generation, enabling the creation of sophisticated, adaptable pipelines for diverse data scenarios. Similarly, Zhang et al. (2024) demonstrated that LLMs could effectively interpret structured inputs and apply past experiences to solve new machine learning tasks. Guo et al. (2024) expanded on this by introducing a data science agent that leverages case-based reasoning; however, it faces challenges when generating solutions from scratch due to its reliance on existing codebases. Schmidt et al. (2024) proposed an iterative approach, where the entire pipeline is generated in one step and refined iteratively through incremental modifications.

Building on these efforts, SELA introduces an agent that integrates the strengths of both approaches—stage-wise planning and iterative refinement—allowing it to autonomously explore and generate machine learning solutions from the ground up. This approach offers greater flexibility and control during the search process, enabling the generation of optimized solutions at each stage.

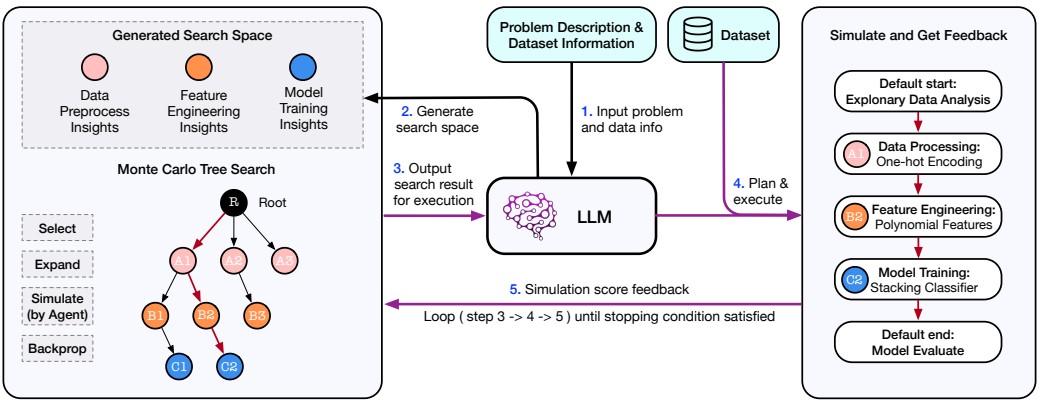

Figure 2: SELA's pipeline operates as follows: The system begins by inputting the problem description and dataset information into the LLM, which generates a search space of potential solutions, encompassing data preprocessing, feature engineering, and model training. The search module, powered by Monte Carlo Tree Search (MCTS), explores this space by selecting, expanding, and simulating potential configurations. The LLM agent then simulates the selected configuration by planning, coding, and executing the experiment. Feedback from the simulation is fed back into the search module, where it is used in the backpropagation step to refine future searches. This iterative process continues until a predefined stopping criterion is met, resulting in an optimized experimental pipeline.

## 3 METHOD

As illustrated in Figure 2, SELA consists of three key components: an LLM-based insight proposer, a search module using MCTS, and an LLM agent as the experiment executor. First, the LLM generates insights from the problem description and dataset, defining a search space. The search module then organizes this space into a tree structure and uses MCTS to explore promising paths. During each cycle, the selected path is passed to the LLM agent, which translates the configuration into an executable pipeline. The agent plans, codes, and executes the experiment, feeding the results back to refine future searches. This iterative process continues until the termination criterion is met. The following sections provide a detailed explanation of each component.

### 3.1 INSIGHT PROPOSAL AND SEARCH SPACE CREATION

To enable SELA to explore a wide range of machine learning strategies, we introduce an insight proposer that generates diverse methods tailored to different stages of the machine learning workflow. Each proposed insight suggests either a single technique or a combination of methods aimed at enhancing performance. For instance, a feature engineering insight might recommend creating interaction features from existing variables, while a model training insight could propose a specific algorithm or suggest running a grid search to improve accuracy.

The insight proposer takes as input the problem description $p$ and dataset information $d$, such as metadata and sample records, and generates $m$ insights $\lambda$ for each stage of the machine learning process using a large language model $M$. These insights are stored in an insight pool, forming a search space $\Lambda$ for SELA to explore. We decompose the machine learning process into five stages: Exploratory Data Analysis ($\tau_1$), Data Preprocessing ($\tau_2$), Feature Engineering ($\tau_3$), Model Training ($\tau_4$), and Model Evaluation ($\tau_5$). For simplicity, we denote the entire set of stages as $T$ and refer to any specific stage as $\tau$.

$$\text{InsightProposer}(p, d, M) \to \Lambda := \{\lambda_i^\tau \mid \tau \in T, i = 1, \dots, m\} \tag{1}$$

### 3.2 PIPELINE EXECUTION AND CODE GENERATION

We employ an LLM agent, referred to as the experiment executor $E$, to conduct each trial by building practical experimental pipelines from natural language requirements. The agent takes two main steps

in this process. First, given an experiment configuration $c$, which is a set of insights provided by the search module (introduced in Section 3.3.2), the experiment executor translates these insights into a detailed plan. This plan consists of a sequence of task instructions $I^{\tau \in T}$ corresponding to each stage of the machine learning process. This step is referred to as $E_{\text{plan}}$.

Next, following the plan, the agent writes and executes code $\sigma^{\tau}$ for each task $\tau$ based on the respective instruction $I^{\tau}$, producing the code $\sigma^{\tau \in T}$ for the full pipeline, along with the final execution score $s$. The complete set of code outputs $\sigma^{\tau \in T}$ is concatenated into a full solution $\sigma_{sol}$ to address the problem. This phase is referred to as $E_{\text{code \& execute}}$.

$$E_{\text{plan}}(p, d, c, M) \rightarrow I^{\tau \in T} \tag{2}$$

$$E_{\text{code \& execute}}(I^{\tau \in T}, D, M) \rightarrow (\sigma^{\tau \in T}, s) \tag{3}$$

## 3.3 Tree Search in Machine Learning Experiments

In order to systematically explore the different configurations in machine learning experiments, we model the search space as a hierarchical tree. This structure allows us to apply tree search algorithms, where each path through the tree represents a different experiment configuration. Algorithm 1 also provides an overview of this searching process.

### 3.3.1 Experiment Node

To facilitate the exploration of various strategies, we model the proposed search space as a hierarchical tree that is well-suited for applying search algorithms. Each node in the tree, denoted as $x$, represents one insight $\lambda$ in the search space $\Lambda$ and contains the following attributes:

- **Insight** $\lambda(x)$: Represents the specific insight $\lambda_i^{\tau} \in \Lambda$ associated with this node, where $\tau$ denotes the stage of the machine learning pipeline.

- **Depth** $\delta(x)$: Indicates the stage of the machine learning process the node corresponds to (e.g., depth 1 might represent data preprocessing, depth 2 for feature engineering, and depth 3 for model training).

- **Value** $v(x)$: The cumulative score from simulations for this node and all its descendants.

- **Number of Visits** $n_{\text{visits}}(x)$: The total number of simulations conducted for this node and its descendants.

- **Simulation Score** $s(x)$: The score for simulating this node.

- **Solution Code** $\sigma_{\text{sol}}(x)$ The final code produced after the node simulation.

- **Stage Code** $\sigma_{\text{stage}}(x)$: The code generated up to the node's current stage, a part of the solution code

By modeling the search space as a tree, each path from the root to a node $x$ represents an experiment configuration $c(x) = \{\lambda(x_1), \lambda(x_2), \dots, \lambda(x)\} \subset \Lambda$, where $x_1, x_2, \dots, x$ are nodes along the path. The task of finding the optimal solution can therefore be viewed as a path search within the tree, where each path corresponds to a potential configuration of the experiment.

### 3.3.2 Tree Search for ML Experiments

We apply Monte Carlo Tree Search (MCTS) to systematically explore and identify optimal machine learning solutions within our framework. MCTS allows us to efficiently navigate the search space across multiple stages of the machine learning pipeline—from data preprocessing to model selection—by balancing exploration and exploitation.

---

**Algorithm 1** SELA using MCTS

---

**Input:** Problem description $p$, data information $d$, data $D$, LLM $M$, rollout number $k$.

1: $\Lambda \leftarrow \text{InsightProposer}(p, d, M)$
2: Initialize Tree using $\Lambda$
3: **for** $i = 1$ **to** $k$ **do**
4:     node $x \leftarrow \text{select(Tree)}$
5:     $X_{\text{child}} \leftarrow \text{expand(Tree, } x)$
6:     Randomly sample a node $x_{\text{sample}}$ from $X_{\text{child}}$
7:     Retreive experiment configuration $c(x_{\text{sample}})$
8:     $\sigma_{sol}, s \leftarrow \text{simulate}(c(x_{\text{sample}}), p, d, D, M)$
9:     attach the simulation result $\sigma_{sol}, s$ to $x_{\text{sample}}$ for final solution selection
10:    Backpropagate(Tree, $s$)
11: **end for**
12: $x_{\text{dev best}} \leftarrow \underset{x \in \text{Tree}}{\text{argmax}}(s(x))$

**Output:** $\sigma_{sol}(x_{\text{dev best}})$

---

**Algorithm 2** Simulate

---

**Input:** Experiment configuration $c$, problem description $p$, data information $d$, data $D$, LLM $M$.

1: Draft plans $I^{\tau \in T} \leftarrow E_{\text{plan}}(p, d, c, M)$
2: Code and execute sequentially $\sigma^{\tau \in T}, s \leftarrow E_{\text{code \& execute}}(I^{\tau \in T}, D, M)$
3: $\sigma_{sol} \leftarrow \text{concatenate}(\sigma^{\tau \in T})$

**Output:** $\sigma_{sol}, s$

---

The search process involves performing multiple rollouts, which include the steps of selection, expansion, simulation, and backpropagation. We conduct $k$ rollouts to explore various paths, aiming to identify the best solution.

**Selection** At each iteration, we use a modified version of the UCT (Upper Confidence Bound for Trees) algorithm, referred to as UCT-DP (depth-preferred), to select a node from the search tree. Unlike traditional MCTS, where simulations are often performed quickly due to a fixed action space and negligible action time, the context of machine learning tasks presents a different challenge. Processes such as model training introduce significant computational time, making efficient node exploration crucial. Since model selection can heavily influence the overall machine learning performance, we prioritize exploring nodes at greater depths early on.

This modification reduces the need to explore every unvisited node, allowing deeper nodes to be reached in fewer iterations—making the approach better suited for large-scale machine learning experiments. The modified selection algorithm is expressed as:

$$\text{UCT-DP}(x) = \frac{v(x)}{n(x)} + \alpha_{\text{explore}} \sqrt{\frac{\ln n_{\text{visits}}(x_{\text{parent}})}{n(x)}} \tag{4}$$

$$n(x) = \begin{cases} \alpha_{\text{unvisted}} & \text{if } n_{\text{visits}}(x) = 0 \\ n_{\text{visits}}(x) & \text{otherwise.} \end{cases} \tag{5}$$

Here, $\alpha_{\text{unvisted}}$ is a constant between 0 and 1 controlling the selection preference for unvisited nodes, balancing between full exploration and computational efficiency. This adjustment allows us to focus more on deeper parts of the tree that are likely to yield better solutions.

**Expansion** During the expansion phase, a set of child nodes $X_{\text{child}}$ at depth $\delta + 1$ are instantiated from the selected node $x$ for potential simulation. Note that a single child node $x_{\text{child}}$ from $x$ inherits the attributes stored in $x$ and possesses $\lambda(x_{\text{child}}) \rightarrow \lambda^{\tau_{\delta+1}}$, an insight of stage $\tau_{\delta+1}$ from the search space.

**Simulation** Once expanded, a node $x_{\text{sample}}$ is randomly sampled from $X_{\text{child}}$ for simulation. The path from root to the sampled node forms a set of insights $c(x_{\text{sample}}) = \{\lambda(x_1), \lambda(x_2), ..., \lambda(x_{\text{sample}})\} \subset \Lambda$, representing the experiment configuration to be simulated, where $x_1, x_2, .., x_{\text{sample}}$ are the nodes along the path. The configuration $c(x_{\text{sample}})$ is then fed to the experimenter $E$ for execution following $E_{\text{plan}}$ and $E_{\text{code \& execute}}$, which produces a simulation score $s$, as illustrated in Section 3.3.1. The score serves as the feedback for back propagation. Algorithm 2 outlines the simulation process.

**Backpropagation** After the simulation concludes, the performance score (e.g., based on the development set) is retrieved and backpropagated through the tree. The score is propagated from the simulated node up to the root, updating each parent node's value and visit count. This allows nodes representing more promising solutions to be prioritized in future rollouts. In addition, the solution code is also backpropagated up to the tree, and it can be processed and saved as stage code depending on the parent node during the update.

Backpropagation ensures that the algorithm learns which paths yield better results, guiding the search toward higher-performing nodes as more rollouts are conducted.

### 3.3.3 Experiment State Saving and Loading

To boost execution efficiency, SELA implements fine-grained code reuse by caching code at the stage level. This caching is done according to each attempted configuration $c$, allowing the framework to reuse as much saved code as possible if the incoming configuration $c_{\text{new}}$ shares any part with existing ones.

Given that LLMs produce non-deterministic outputs, the same instruction can yield different code, leading to greater variance in final performance. To minimize this variance and reduce token usage during execution, SELA saves and loads the stage code for each node. Whenever a node is chosen for execution, the experimenter reruns the saved stage code, ensuring consistency before progressing to the next stage. This approach effectively conserves resources while maintaining robust performance across stages. In Appendix D, we examine the cost efficiency of this state-saving and loading mechanism.

## 4 Experiments

### 4.1 Experimental Setup

**Datasets** We evaluate SELA alongside several baselines on 20 datasets, which include 13 classification tasks and 7 regression tasks from the AutoML Benchmark (AMLB) (Gijsbers et al., 2024) and Kaggle Competitions.

Table 3 provides detailed information on the datasets used. All datasets are split into training, validation, and test sets with a 6:2:2 ratio. Each framework utilizes the training and validation sets to train models and makes predictions on the test set labels.

**Evaluation Metrics** For the AMLB datasets, we use the default target column provided by OpenML. For Kaggle competition datasets, we rely on the target column specified in the competition description. Performance is measured using root mean squared error (RMSE) for regression tasks, F1 score for binary classification, and F1-weighted score for multi-class classification. To ensure comparability across datasets with varying metrics, we introduce a normalized score (NS), which intends to map RMSE into a range from 0 to 1.

$$\text{NS}(s_{\text{raw}}) = \begin{cases} \frac{1}{1+\log{(1+s_{\text{raw}})}} & \text{if the metric is RMSE.} \\ s_{\text{raw}} & \text{otherwise.} \end{cases} \tag{6}$$

Here, $s_{raw}$ represents the raw score before normalization. To evaluate SELA against other frameworks, we employ three key metrics: average Normalized Score (NS), average rank, and average best rank. The average rank is calculated by considering all rankings of a method across datasets, while the average best rank focuses on the method's best performance in each dataset. We also want

to quantify how other baselines perform relative to SELA. The "Rescaled NS" is defined as:

$$\text{Rescaled NS}(f) = \frac{\text{NS}_f}{\text{NS}_{\text{SELA}}} \tag{7}$$

where $f$ represents the baseline method being compared to SELA.

**Baselines** We compare SELA with several baseline methods, including Data Interpreter (Hong et al., 2024), AIDE (Schmidt et al., 2024), AutoGluon (Erickson et al., 2020), and AutoSklearn (Feurer et al., 2015; 2020).

For LLM-based methods (SELA, Data Interpreter (DI), and AIDE), we use the same initial task prompt, which includes the dataset name, target column, and evaluation metric. Given that DeepSeek v2.5 (DeepSeek-AI, 2024) is an open-source large language model with robust coding capabilities and a relatively low token cost, we selected it as the base LLM for our experiments. To promote a moderate level of diversity in the model's outputs, we set the temperature parameter to 0.5. AIDE performs 10 iterations per execution, while SELA uses DI as the experimenter and completes 10 rollouts per execution.

Each method, except for AutoGluon, is run three times for each dataset. AutoGluon, being deterministic, is run only once with its default settings. AutoSklearn is also run with default settings, limited to 600 seconds per task.

| Method | Wins | Losses | Top 1 | Avg. NS % ↑ | Avg. Best NS % ↑ | Avg. Rank ↓ | Avg. Best Rank ↓ |
|---|---|---|---|---|---|---|---|
| AutoGluon | 7 | 13 | 4 | 53.2 | 53.2 | **4.4** | 4.4 |
| AutoSklearn | 5 | 15 | 5 | 46.1 | 47.5 | 7.6 | 6.1 |
| AIDE | 5 | 15 | 2 | 47.1 | 51.8 | 7.8 | 5.3 |
| Data Interpreter | 4 | 16 | 2 | 47.4 | 50.2 | 8.8 | 6.4 |
| SELA | - | - | **7** | **53.3** | **54.7** | 4.8 | **2.7** |

Table 1: Results of each AutoML framework on 20 tabular datasets. The "Wins" column indicates the number of datasets where the method outperforms SELA, while "Losses" shows the number of datasets where the method underperforms. The "Top 1" column represents the number of datasets where the method produces the best predictions across methods.

## 4.2 RESULTS

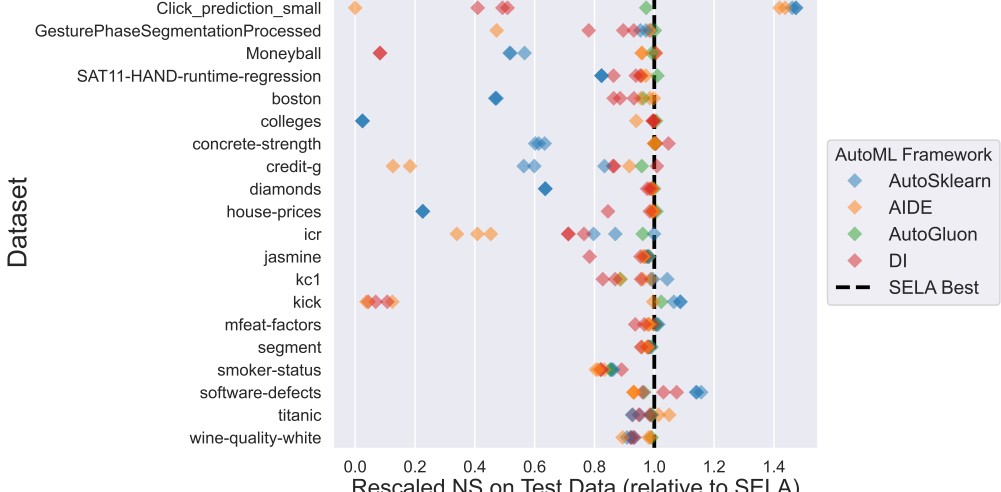

Figure 3: Rescaled NS of AutoML frameworks relative to SELA on tabular datasets. Points to the left of the vertical line indicate poorer predictions compared to SELA. Notably, SELA often occupies a leading position across the datasets.

As shown in Table 1, SELA achieves the highest average Normalized Score (NS) and average best rank among all frameworks. Notably, SELA excels in producing the highest number of top predictions, as indicated in the "Top 1" column across all datasets. Furthermore, the "Losses" column reveals that each competing method falls short against SELA, losing in 65-80% of the datasets.

Interestingly, AutoGluon exhibits a marginally higher average rank than SELA. This slight discrepancy may be attributed to the inherent randomness in LLMs and model training processes, which can influence the exploration of machine learning solutions. However, SELA's higher average NS suggests that even when it produces solutions with lower ranks, their test scores remain competitive and close to the best solutions.

The two agent-based methods exhibit relatively lower performance. The first method, DI, struggles to enhance its score with multiple attempts due to its inability to refine its solution after completing a machine learning task. The second method, AIDE, lacks a stage-specific planning module, which hinders its capacity to improve results after a series of greedy experiments. These limitations likely account for their weaker performance.

Figure 3 further corroborates SELA's effectiveness, revealing that its best solutions frequently occupy leading positions across various datasets. This visual representation exhibits the method's consistent high performance and adaptability across different ML datasets. We also include a detailed results of each method in Appendix C.

## 4.3 ABLATION STUDY

For the rest of the study, we employ a subset of datasets to evaluate SELA under various settings. Our selection process involves choosing the first two datasets alphabetically for each machine learning task. Specifically, we use boston, colleges, credit-g, Click_prediction_small, GesturePhaseSegmentationProcessed, and mfeat-factors to conduct the ablation study.

|  | DI | SELA (Random Search) | SELA (MCTS) |
|---|---|---|---|
| Avg. NS ↑ | 56.4 | 58.6 | **60.9** |
| Avg. Best NS ↑ | 59.0 | 61.4 | **62.4** |
| Avg. Rank ↓ | 6.9 | 4.8 | **3.3** |
| Avg. Best Rank ↓ | 4.8 | 2.8 | **1.5** |

Table 2: Performance results for each search setting on the chosen datasets. SELA with MCTS consistently surpasses SELA with Random Search.

**Effectiveness of Search**  To evaluate the effectiveness of Monte Carlo Tree Search (MCTS) in improving the solution search process, we conducted an ablation study. In this study, we compared the performance of our method using MCTS against a variant that randomly samples insights from each stage's insight pool. As shown in Table 2, the MCTS version achieves a higher average normalized score across datasets and a better overall ranking compared to the random sampling approach. Moreover, even the random sampling variant of our method outperforms DI, the base experimenter. This suggests the presence of an appropriate search space and an experiment agenda is vital for improving a machine learning agent. Our insight proposer generates relevant and useful insights, facilitating such improvement, regardless of the selection method.

**SELA's performance with different LLMs**  To evaluate the robustness of our framework, we conduct experiments using different Large Language Models (LLMs). Specifically, we compare the performance of SELA with `Claude-3.5-Sonnet` (Anthropic, 2024) and `GPT-4o` (OpenAI, 2024) against `DeepSeek V2.5` which we primarily use for evaluation. This comparison enables us to assess how the choice of LLM affects the overall effectiveness of our approach.

As Figure 4 shown, SELA delivers similar results across different LLMs, indicating its flexibility to be executed with various models depending on user preference and availability. We also report the numeric results in Appendix C.2.

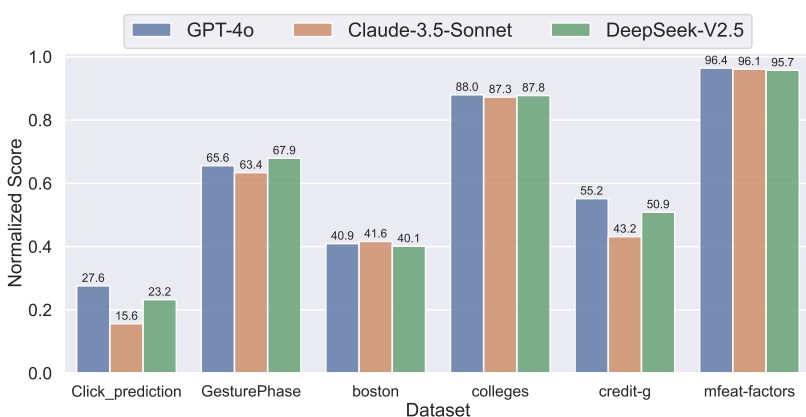

Figure 4: Comparison of Normalized Scores between different base LLMs.

## 5  CONCLUSION

In this paper, we introduced SELA, a novel framework that integrates LLM-based agents with Monte Carlo Tree Search (MCTS) to automate machine learning workflows. Our experimental results, conducted on 20 machine learning datasets, demonstrate SELA's effectiveness and highlight its distinct advantages over both traditional AutoML frameworks and existing LLM-based approaches. The proposed methodology is not limited to machine learning but could be adapted to a wide range of sequential decision-making problems, provided they can be represented as tree structures with scalar rewards derived from their leaf nodes. Looking ahead, future work could explore extending this framework to other domains, including software engineering, scientific discovery, game playing, and robotics. Furthermore, improving the efficiency and scalability of the tree search process for larger solution spaces remains an important area for investigation. Another promising direction is developing techniques to provide interpretable explanations of the search process and solution rationale, enhancing the transparency and trustworthiness of the system. SELA represents a significant advancement in automated machine learning, demonstrating the potential of combining traditional search algorithms with the flexibility of LLMs.

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

# A  DATASETS

Table 3 outlines the detailed information of the datasets used for evaluation.

| Dataset name | # Features | # Rows | # Classes | Task Type | Metric | Source |
|---|---|---|---|---|---|---|
| boston | 14 | 506 | N/A | Regression | RMSE | OpenML (Dataset ID: 531) |
| colleges | 48 | 7063 | N/A | Regression | RMSE | OpenML (Dataset ID: 42727) |
| concrete-strength | 9 | 4866 | N/A | Regression | RMSE | Kaggle (playground-series-s3e9) |
| diamonds | 10 | 53940 | N/A | Regression | RMSE | OpenML (Dataset ID: 42225) |
| house-prices | 81 | 1460 | N/A | Regression | RMSE | Kaggle (house-prices-advanced-regression-techniques) |
| Moneyball | 15 | 1232 | N/A | Regression | RMSE | OpenML (Dataset ID: 41021) |
| SAT11-HAND-runtime-regression | 118 | 4440 | N/A | Regression | RMSE | OpenML (Dataset ID: 41980) |
| credit-g | 21 | 1000 | 2 | Classification | F1 | OpenML (Dataset ID: 31) |
| Click_prediction_small | 12 | 39948 | 2 | Classification | F1 | OpenML (Dataset ID: 42733) |
| icr | 58 | 617 | 2 | Classification | F1 | Kaggle (icr-identify-age-related-conditions) |
| jasmine | 145 | 2984 | 2 | Classification | F1 | OpenML (Dataset ID: 41143) |
| kc1 | 21 | 2109 | 2 | Classification | F1 | OpenML (Dataset ID: 1067) |
| kick | 33 | 72983 | 2 | Classification | F1 | OpenML (Dataset ID: 41162) |
| smoker-status | 23 | 143330 | 2 | Classification | F1 | Kaggle (playground-series-s3e24) |
| software-defects | 22 | 91586 | 2 | Classification | F1 | Kaggle (playground-series-s3e23) |
| titanic | 12 | 891 | 2 | Classification | F1 | Kaggle (titanic) |
| GesturePhaseSegmentationProcessed | 33 | 9873 | 5 | Multiclass | F1-weighted | OpenML (Dataset ID: 4538) |
| mfeat-factors | 217 | 2000 | 10 | Multiclass | F1-weighted | OpenML (Dataset ID: 12) |
| segment | 20 | 2310 | 7 | Multiclass | F1-weighted | OpenML (Dataset ID: 40984) |
| wine-quality-white | 12 | 4898 | 7 | Multiclass | F1-weighted | OpenML (Dataset ID: 40498) |

Table 3: Summary of the machine learning datasets used in the experiments. OpenML datasets can be accessed using their respective dataset IDs. The Kaggle datasets are available at https://www.kaggle.com/competitions/{source}.

# B    PROMPTS

## B.1    TASK PROMPT

All LLM-based methods start by receiving the same base requirement prompt at the beginning of the task. The prompt specifies the dataset's name, the target label column, the evaluation metric to be used, and the dataset's file path. Furthermore, the prompt include a path to a text file containing the dataset's metadata.

```
1  TASK_PROMPT = """
2  # User requirement
3  This is a {datasetname} dataset.
4  Your goal is to predict the target column `{target_col}`.
5  Perform data analysis, data preprocessing, feature engineering, and modeling to predict the
       target. Report {metric} on the eval data. Do not plot or make any visualizations.
6
7  # Data dir
8  train set (with labels): {train_path}
9  dev set (with labels): {dev_path}
10 test set (without labels): {test_path}
11 dataset description: {data_info_path}
12 (During EDA, you can use this file
13 to get additional information about the dataset)
14 """
```

Since AIDE automatically splits the training set into a new train set and a dev set, we combine the original train and dev sets and provide them as input to AIDE. In both setups, the frameworks have access to the labels for both the train and dev sets. Therefore, we believe this subtle difference does not affect the fairness of the comparison.

## B.2    INSTRUCTION PROMPT

The instruction prompt would direct the framework to save the final prediction file for evaluation.

```
1  DI_INSTRUCTION = """
2  ## Attention
3  1. Please do not leak the target label in any form during training.
4  2. Test set does not have the target column.
5  3. When conducting data exploration or analysis, print out the results of your findings.
6  4. You should perform transformations on train, dev, and test sets at the same time (it's a
       good idea to define functions for this and avoid code repetition).
7  5. When scaling or transforming features, make sure the target column is not included.
8  6. You could utilize dev set to validate and improve model training. {special_instruction}
9
10 ## Saving Dev and Test Predictions
11 1. Save the prediction results of BOTH the dev set and test set in `dev_predictions.csv` and `
       test_predictions.csv` respectively in the output directory.
12 - Both files should contain a single column named `target` with the predicted values.
13 2. Make sure the prediction results are in the same format as the target column in the
       training set.
14 - For instance, if the target column is categorical, the prediction results should be
       categorical as well.
15
16 ## Output Performance
17 Print the train and dev set performance in the last step.
18
19 # Output dir
20 {output_dir}
21 """
```

## B.3 INSIGHT PROPOSAL PROMPT

Insight Proposer uses this prompt to generate a search space of insights for different stages of the machine learning task.

```
DATASET_INSIGHT_PROMPT = """
# Dataset Description
{dataset}

# Dataset Metadata
{metadata}

# Dataset Head
{head}

# Instruction
Propose insights to help improve the performance of the model on this dataset.
The insights should be proposed based on the dataset description with different task types.
Each task type should have at least 5 insights.
Make sure each method is diverse enough and can be implemented separately.
Be specific about models' choices, ensemble and tuning techniques, and preprocessing & feature
    engineering techniques.

# Format
```json
[
    {{
        "task_type": "EDA",
        "insights": [
            "insight1",
            "insight2",
            "insight3",
            ...
            "insightN"
        ]
    }},
    {{
        "task_type": "Data Preprocessing",
        "insights": [
            "insight1",
            "insight2",
            "insight3",
            ...
            "insightN"
        ]
    }},
    {{
        "task_type": "Feature Engineering",
        "insights": [
            "insight1",
            "insight2",
            "insight3",
            ...
            "insightN"
        ]
    }},
    {{
        "task_type": "Model Training",
        "insights": [
            "insight1",
            "insight2",
            "insight3",
            ...
            "insightN"
        ]
    }}
]
```
"""
```

## C RESULTS

### C.1 MAIN RESULTS

| Dataset | AutoGluon | | AutoSklearn | | AIDE | | DI | | SELA | |
|---|---|---|---|---|---|---|---|---|---|---|
| | Avg. | Best | Avg. | Best | Avg. | Best | Avg. | Best | Avg. | Best |
| Click_prediction_small | 7 | 7 | 2 | 1 | 7.3 | 4 | 11 | 10 | 7.7 | 6 |
| GesturePhaseSegmentationProcessed | 1 | 1 | 6.3 | 3 | 7.3 | 4 | 11 | 10 | 5.3 | 2 |
| Moneyball | 4 | 4 | 10 | 9 | 4 | 1 | 9 | 2 | 6 | 3 |
| SAT11-HAND-runtime-regression | 1 | 1 | 12 | 11 | 5.3 | 3 | 9 | 8 | 3.7 | 2 |
| boston | 5 | 5 | 12 | 11 | 3.7 | 2 | 9 | 8 | 4 | 1 |
| colleges | 1 | 1 | 12 | 11 | 6 | 2 | 8 | 7 | 4 | 3 |
| concrete-strength | 5 | 5 | 12 | 11 | 6.3 | 4 | 2 | 1 | 8.3 | 6 |
| credit-g | 4 | 4 | 10 | 9 | 10 | 5 | 5.3 | 1 | 3.7 | 2 |
| diamonds | 2 | 2 | 12 | 11 | 6 | 4 | 8.7 | 7 | 3 | 1 |
| house-prices | 1 | 1 | 12 | 11 | 6.7 | 5 | 7.3 | 3 | 4 | 2 |
| icr | 5 | 5 | 5.3 | 3 | 12 | 11 | 9 | 8 | 2.3 | 1 |
| jasmine | 7 | 7 | 6 | 4 | 8.7 | 5 | 11.3 | 9 | 2 | 1 |
| kc1 | 10 | 10 | 2.7 | 1 | 8 | 5 | 11.3 | 9 | 5 | 2 |
| kick | 4 | 4 | 2 | 1 | 9.3 | 6 | 11 | 10 | 6.7 | 5 |
| mfeat-factors | 4 | 4 | 2 | 1 | 10 | 9 | 10.3 | 6 | 6.7 | 5 |
| segment | 3 | 3 | 6.3 | 5 | 11 | 10 | 9.7 | 7 | 2.3 | 1 |
| smoker-status | 7 | 7 | 4.7 | 3 | 11.3 | 9 | 7.7 | 2 | 4.3 | 1 |
| software-defects | 8 | 8 | 2 | 1 | 12 | 11 | 6 | 4 | 7.7 | 6 |
| titanic | 7 | 7 | 9.7 | 6 | 2.7 | 1 | 10.3 | 8 | 5.3 | 3 |
| wine-quality-white | 2 | 2 | 10 | 8 | 7.3 | 4 | 9 | 7 | 3.3 | 1 |
| Overall Rank ↓ | **4.4** | 4.4 | 7.6 | 6.1 | 7.8 | 5.3 | 8.8 | 6.4 | 4.8 | **2.7** |

Table 4: Methods' ranking for each tabular dataset

| Dataset | AutoGluon | | AutoSklearn | | AIDE | | DI | | SELA | |
|---|---|---|---|---|---|---|---|---|---|---|
| | Avg. | Best | Avg. | Best | Avg. | Best | Avg. | Best | Avg. | Best |
| Click_prediction_small | 26.6 | 26.6 | 40.2 | 40.3 | 26.1 | 39.4 | 12.9 | 13.9 | 23.2 | 27.4 |
| GesturePhaseSegmentationProcessed | 69.3 | 69.3 | 67.2 | 68.4 | 56.3 | 68.1 | 60.1 | 64.4 | 67.9 | 69.2 |
| Moneyball | 24.3 | 24.3 | 13.1 | 13.8 | 23.8 | 24.6 | 9.5 | 24.5 | 21.9 | 24.5 |
| SAT11-HAND-runtime-regression | 12.6 | 12.6 | 10.3 | 10.3 | 12.0 | 12.1 | 11.4 | 11.9 | 12.2 | 12.5 |
| boston | 39.8 | 39.8 | 19.5 | 19.6 | 40.5 | 41.3 | 37.0 | 38.6 | 40.1 | 41.4 |
| colleges | 88.3 | 88.3 | 2.1 | 2.1 | 86.0 | 87.8 | 87.5 | 87.7 | 87.8 | 87.8 |
| concrete-strength | 28.3 | 28.3 | 17.4 | 17.9 | 28.3 | 28.3 | 28.8 | 29.6 | 28.2 | 28.2 |
| credit-g | 50.5 | 50.5 | 35.1 | 44.0 | 21.6 | 48.4 | 48.1 | 53.2 | 50.9 | 52.7 |
| diamonds | 13.8 | 13.8 | 8.7 | 8.7 | 13.7 | 13.7 | 13.5 | 13.6 | 13.7 | 13.8 |
| house-prices | 9.0 | 9.0 | 2.0 | 2.0 | 8.9 | 8.9 | 8.5 | 9.0 | 8.9 | 9.0 |
| icr | 76.2 | 76.2 | 70.4 | 79.2 | 31.7 | 35.9 | 57.8 | 60.6 | 78.7 | 79.2 |
| jasmine | 84.3 | 84.3 | 84.4 | 84.7 | 83.6 | 84.6 | 77.8 | 83.5 | 85.4 | 86.2 |
| kc1 | 38.3 | 38.3 | 43.5 | 45.0 | 40.8 | 42.6 | 38.1 | 41.2 | 42.2 | 43.1 |
| kick | 39.6 | 39.6 | 41.8 | 42.1 | 14.9 | 38.6 | 2.8 | 4.2 | 35.9 | 38.7 |
| mfeat-factors | 96.7 | 96.7 | 97.1 | 97.5 | 94.4 | 94.5 | 93.0 | 96.0 | 95.7 | 96.2 |
| segment | 93.5 | 93.5 | 92.7 | 93.1 | 91.7 | 92.2 | 91.7 | 92.6 | 93.8 | 94.4 |
| smoker-status | 78.0 | 78.0 | 78.6 | 78.9 | 74.8 | 76.3 | 77.3 | 81.5 | 82.4 | 91.5 |
| software-defects | 51.5 | 51.5 | 61.1 | 61.7 | 49.7 | 49.8 | 54.5 | 57.3 | 52.2 | 53.3 |
| titanic | 78.9 | 78.9 | 76.2 | 78.9 | 81.2 | 83.7 | 76.0 | 78.5 | 78.8 | 79.7 |
| wine-quality-white | 65.4 | 65.4 | 60.7 | 61.4 | 62.9 | 65.1 | 61.2 | 61.6 | 65.3 | 66.0 |
| Overall NS % ↑ | 53.2 | 53.2 | 46.1 | 47.5 | 45.5 | 51.8 | 47.4 | 50.2 | **53.3** | **54.7** |

Table 5: Methods' NS % for each tabular dataset

## C.2    PERFORMANCE USING DIFFERENT LLMs

|                  | GPT-4o | Claude 3.5 Sonnet | DeepSeek V2.5 |
|------------------|--------|-------------------|---------------|
| Avg. NS ↑        | **62.3** | 57.9            | 60.9          |
| Avg. Best NS ↑   | **65.5** | 59.2            | 62.4          |
| Avg. Rank ↓      | **3.7**  | 6.3             | 5.0           |
| Avg. Best Rank ↓ | **1.5**  | 4.8             | 3.2           |

Table 6: Results of SELA with different base LLMs on the selected tabular datasets.

# D    COST-EFFECTIVENESS ANALYSIS

We conduct multiple trials of execution of each method to estimate the average running cost for the LLM-based baselines. As shown in Table 7, all methods incur relatively low costs to complete a single machine learning task. Among these, AIDE exhibits the lowest execution cost, due to the lack of stage-wise planning, resulting in fewer token generations compared to the other approaches. Additionally, SELA, which employs Data Interpreter as its base experimenter, is less costly than Data Interpreter itself. This efficiency is largely due to SELA's state-saving and loading mechanism, which reduces the generation of repeated tasks and code.

|  | Cost per ML task ($) |
| --- | --- |
| Data Interpreter ($k$=10) | 0.07 |
| AIDE ($k$=10) | 0.01 |
| SELA ($k$=10) | 0.05 |

Table 7: Estimated costs of agent-based frameworks utilizing DeepSeekV2.5 on a single machine learning dataset over $k$ iterations/rollouts.

# E CASE STUDY

## E.1 MCTS PROCESS OVERVIEW

```
Number of simulations: 10
[Node 0]
Plans:
1. Perform exploratory data analysis on the train and dev datasets
2. Preprocess the train, dev, and test datasets
3. Perform feature engineering on the train, dev, and test datasets
4. Train multiple models and evaluate their performance
5. Train a weighted ensemble model using the best performing models
6. Evaluate the ensemble model on the dev set and save predictions
7. Generate predictions for the test set and save them
Simulated: True
Score: avg score: 0.6150206840685731, simulated score: {'train_score': 1.0, 'dev_score':
    0.6855841857240594, 'test_score': 0.6814818772150697, 'score': 0.6855841857240594},
    Visits: 10

    [Node 0-0]
    Plans:
    3. Perform feature engineering on the train, dev, and test datasets by creating new
        features that calculate the magnitude of the vectorial velocities and accelerations
        to capture the overall movement intensity.
    Simulated: True
    Score: avg score: 0.6507249985568175, simulated score: {'train_score': 0.982920964830782,
        'dev_score': 0.6420233166755841, 'test_score': 0.647550336228104, 'score':
        0.6420233166755841}, Visits: 2

        [Node 0-0-0]
        Plans:
        4. Train a Random Forest classifier to leverage its ability to handle
            high-dimensional data and capture non-linear relationships, and evaluate its
            performance
        Simulated: False
        Score: avg score: 0, simulated score: {}, Visits: 0

        [Node 0-0-1]
        Plans:
        4. Train multiple models, including a Support Vector Machine (SVM) with a radial
            basis function (RBF) kernel, and evaluate their performance.
        Simulated: False
        Score: avg score: 0, simulated score: {}, Visits: 0

        [Node 0-0-2]
        Plans:
        4. Implement a Neural Network with multiple layers to capture the hierarchical
            patterns in the data and evaluate its performance
        Simulated: True
        Score: avg score: 0.6594266804380511, simulated score: {'train_score': 1.0,
            'dev_score': 0.6594266804380511, 'test_score': 0.6702614538699305, 'score':
            0.6594266804380511}, Visits: 1

        [Node 0-0-3]
        Plans:
        4. Train multiple models, apply an ensemble method like Gradient Boosting to combine
            them, and evaluate their performance
        Simulated: False
        Score: avg score: 0, simulated score: {}, Visits: 0

        [Node 0-0-4]
        Plans:
        4. Train multiple models, perform hyperparameter tuning using Grid Search or Random
            Search, and evaluate their performance
        Simulated: False
        Score: avg score: 0, simulated score: {}, Visits: 0

    [Node 0-1]
    Plans:
    3. Perform feature engineering on the train, dev, and test datasets by generating
        time-based features, such as the difference between consecutive frames, to capture
        the rate of change in movements.
    Simulated: True
    Score: avg score: 0.6464940718972336, simulated score: {'train_score': 1.0, 'dev_score':
        0.5985614604756948, 'test_score': 0.5857379626419719, 'score': 0.5985614604756948},
        Visits: 2

        [Node 0-1-0]
        Plans:
```

```
  58        4. Train a Random Forest classifier to leverage its ability to handle
                 high-dimensional data and capture non-linear relationships
  59        Simulated: False
  60        Score: avg score: 0, simulated score: {}, Visits: 0
  61
  62        [Node 0-1-1]
  63        Plans:
  64        4. Train multiple models, including a Support Vector Machine (SVM) with a radial
                 basis function (RBF) kernel, and evaluate their performance to model the complex
                 decision boundaries between different gesture phases.
  65        Simulated: True
  66        Score: avg score: 0.6944266833187726, simulated score: {'train_score': 1.0,
                 'dev_score': 0.6944266833187726, 'test_score': 0.6928451194338062, 'score':
                 0.6944266833187726}, Visits: 1
  67
  68        [Node 0-1-2]
  69        Plans:
  70        4. Implement a Neural Network with multiple layers to capture the hierarchical
                 patterns in the data and evaluate its performance
  71        Simulated: False
  72        Score: avg score: 0, simulated score: {}, Visits: 0
  73
  74        [Node 0-1-3]
  75        Plans:
  76        4. Train multiple models, apply an ensemble method like Gradient Boosting to combine
                 them, and evaluate their performance
  77        Simulated: False
  78        Score: avg score: 0, simulated score: {}, Visits: 0
  79
  80        [Node 0-1-4]
  81        Plans:
  82        4. Train multiple models and perform hyperparameter tuning using techniques like Grid
                 Search or Random Search to optimize and evaluate their performance.
  83        Simulated: False
  84        Score: avg score: 0, simulated score: {}, Visits: 0
  85
  86    [Node 0-2]
  87    Plans:
  88    3. Perform feature engineering on the train, dev, and test datasets by creating features
             that represent the spatial relationships between different body parts, such as the
             distance between the hands and the head.
  89    Simulated: True
  90    Score: avg score: 0.6296836159165489, simulated score: {'train_score':
             0.7619969104124632, 'dev_score': 0.5997286931710517, 'test_score':
             0.604077566134264, 'score': 0.5997286931710517}, Visits: 3
  91
  92        [Node 0-2-0]
  93        Plans:
  94        4. Train a Random Forest classifier to leverage its ability to handle
                 high-dimensional data and capture non-linear relationships, and evaluate its
                 performance
  95        Simulated: False
  96        Score: avg score: 0, simulated score: {}, Visits: 0
  97
  98        [Node 0-2-1]
  99        Plans:
 100        4. Train multiple models, including a Support Vector Machine (SVM) with a radial
                 basis function (RBF) kernel, and evaluate their performance to model the complex
                 decision boundaries between different gesture phases.
 101        Simulated: True
 102        Score: avg score: 0.6446610772892973, simulated score: {'train_score':
                 0.9952809245924918, 'dev_score': 0.6372459669415207, 'test_score':
                 0.6423549137767338, 'score': 0.6372459669415207}, Visits: 2
 103
 104            [Node 0-2-1-0]
 105            Plans:
 106            5. Train a weighted ensemble model using the best performing models from task 4
 107            Simulated: False
 108            Score: avg score: 0, simulated score: {}, Visits: 0
 109
 110            [Node 0-2-1-1]
 111            Plans:
 112            5. Using the models that performed best in task 4, train a weighted ensemble
                     model to improve overall performance.
 113            Simulated: False
 114            Score: avg score: 0, simulated score: {}, Visits: 0
 115
 116            [Node 0-2-1-2]
 117            Plans:
```

```
         5. Develop a weighted ensemble model by integrating the top-performing models
             from task 4, ensuring to evaluate and adjust the weights for optimal
             performance.
         Simulated: True
         Score: avg score: 0.6520761876370741, simulated score: {'train_score': 1.0,
             'dev_score': 0.6520761876370741, 'test_score': 0.6563435152603494, 'score':
             0.6520761876370741}, Visits: 1

         [Node 0-2-1-3]
         Plans:
         5. Train a weighted ensemble model by combining the predictions of the
             top-performing models from task 4 to improve overall performance.
         Simulated: False
         Score: avg score: 0, simulated score: {}, Visits: 0

         [Node 0-2-1-4]
         Plans:
         5. Develop a weighted ensemble model by combining the top-performing models from
             task 4, ensuring to optimize the weights for improved performance.
         Simulated: False
         Score: avg score: 0, simulated score: {}, Visits: 0

     [Node 0-2-2]
     Plans:
     4. Implement a Neural Network with multiple layers to capture the hierarchical
         patterns in the data and evaluate its performance
     Simulated: False
     Score: avg score: 0, simulated score: {}, Visits: 0

     [Node 0-2-3]
     Plans:
     4. Train multiple models, apply an ensemble method like Gradient Boosting to combine
         them, and evaluate their performance
     Simulated: False
     Score: avg score: 0, simulated score: {}, Visits: 0

     [Node 0-2-4]
     Plans:
     4. Perform hyperparameter tuning using Grid Search or Random Search to train multiple
         models and evaluate their performance
     Simulated: False
     Score: avg score: 0, simulated score: {}, Visits: 0

[Node 0-3]
Plans:
3. Apply feature selection techniques such as Recursive Feature Elimination (RFE) or
    SelectKBest to identify and retain the most important features in the train, dev,
    and test datasets.
Simulated: True
Score: avg score: 0.49056683315196203, simulated score: {'train_score':
    0.9988177730410426, 'dev_score': 0.51620611302976, 'test_score': 0.525989891002361,
    'score': 0.51620611302976}, Visits: 2

     [Node 0-3-0]
     Plans:
     4. Train a Random Forest classifier to leverage its ability to handle
         high-dimensional data and capture non-linear relationships, and evaluate its
         performance.
     Simulated: False
     Score: avg score: 0, simulated score: {}, Visits: 0

     [Node 0-3-1]
     Plans:
     4. Train multiple models, including a Support Vector Machine (SVM) with a radial
         basis function (RBF) kernel, and evaluate their performance to model the complex
         decision boundaries between different gesture phases.
     Simulated: True
     Score: avg score: 0.4649275532741641, simulated score: {'train_score':
         0.7299159411193588, 'dev_score': 0.4649275532741641, 'test_score':
         0.4631598897487413, 'score': 0.4649275532741641}, Visits: 1

     [Node 0-3-2]
     Plans:
     4. Implement and train a Neural Network with multiple layers to capture hierarchical
         patterns in the data and evaluate its performance
     Simulated: False
     Score: avg score: 0, simulated score: {}, Visits: 0

     [Node 0-3-3]
     Plans:
```

```
178      4. Train multiple models, apply an ensemble method like Gradient Boosting to combine
            them, and evaluate their performance
179      Simulated: False
180      Score: avg score: 0, simulated score: {}, Visits: 0
181
182      [Node 0-3-4]
183      Plans:
184      4. Train multiple models, perform hyperparameter tuning using techniques like Grid
            Search or Random Search, and evaluate their performance
185      Simulated: False
186      Score: avg score: 0, simulated score: {}, Visits: 0
187
188    [Node 0-4]
189    Plans:
190    3. Create interaction features by combining existing features, such as the product of
          velocity and acceleration, to capture complex relationships in the train, dev, and
          test datasets
191    Simulated: False
192    Score: avg score: 0, simulated score: {}, Visits: 0
193
194  Generated 29 unique codes.
195  Best node: 0-1-1, score: {'train_score': 1.0, 'dev_score': 0.6944266833187726, 'test_score':
        0.6928451194338062, 'score': 0.6944266833187726}
196  Dev best node: 0-1-1, score: {'train_score': 1.0, 'dev_score': 0.6944266833187726,
        'test_score': 0.6928451194338062, 'score': 0.6944266833187726}
```

The MCTS process in this case study consists of a structured exploration of the machine learning pipeline, executed in the following steps:

**Step 1: Initialization (Node 0)**
The process begins by defining high-level tasks, such as data analysis, data pre-processing, feature engineering, and model training. These general steps establish the overall framework for the machine learning workflow.

**Step 2: Feature Engineering Exploration (Selection and Expansion)**
MCTS then explores specific feature engineering techniques. For instance, Node 0-0 introduces features like the magnitude of vectorial velocities, Node 0-1 generates time-based features, and Node 0-2 creates spatial relationship features between body parts. These feature engineer methods aim to improve data representation, which is crucial for enhancing model accuracy.

**Step 3: Model Training (Expansion)**
At this point, the process tests various machine learning models. For example, Node 0-1-1 applies a Support Vector Machine (SVM) with a radial basis function (RBF) kernel, while Node 0-0-2 evaluates a Neural Network. The models are trained and evaluated based on performance across training, development, and test datasets.

**Step 4: Performance Evaluation (Simulation)**
Each node is scored based on model performance. MCTS retains and further explores the best-performing nodes, using prior successful results to guide the search for improved solutions.

**Step 5: Nodes Update (Backpropagation)**
After the simulation, the performance score is retrieved and backpropagated through the tree. For example, after simulating Node 0-1-1, MCTS backpropagates the result up the tree, updating parent nodes like Node 0-1 and Node 0.

**Step 6: Best Model Selection**
In the final step, MCTS selects the best-performing solution. In this case, Node 0-1-1, using the SVM with RBF kernel, achieved the highest scores across datasets, effectively combining feature engineering and model selection to optimize the machine learning pipeline.

