# OpenReview forum: "SELA: Tree-Search Enhanced LLM Agents for Automated Machine Learning"
_ICLR.cc/2025/Conference — Submitted to ICLR 2025_

### Official Review · Reviewer_epEJ · 2024-11-01

**Soundness:** 3
**Presentation:** 3
**Contribution:** 2
**Rating:** 5
**Confidence:** 4

**Summary:**

The paper introduces SELA, a novel framework that integrates Monte Carlo Tree Search (MCTS) with Large Language Model (LLM) agents to enhance the process of Automated Machine Learning (AutoML). The authors claim that SELA improves the exploration and optimization of machine learning pipelines by leveraging MCTS to systematically explore potential configurations. The framework is evaluated on 20 datasets, demonstrating a win rate of 65% to 80% against traditional and agent-based AutoML methods.

**Strengths:**

- The integration of MCTS with LLM agents is a creative approach to address the limitations of existing AutoML frameworks. By using MCTS, SELA aims to mimic the iterative and feedback-driven problem-solving approach of human experts, which is a commendable attempt to enhance the adaptability and effectiveness of AutoML systems.
- The experimental results indicate that SELA outperforms several baseline methods across a range of datasets. This suggests that the proposed method has potential advantages in terms of flexibility and performance, offering a fresh perspective on tackling complex machine learning challenges.
- The paper provides a detailed description of the SELA framework, including its components such as the insight proposer, experiment executor, and the MCTS-based search module. This thorough explanation helps in understanding the workflow and the role of each component in the overall system.

**Weaknesses:**

- While the integration of MCTS with LLM agents is an interesting approach, the innovation appears limited. The application of MCTS in this context is relatively straightforward and does not introduce significant novel techniques beyond existing LLM-agent frameworks like Data Interpreter. The paper could benefit from a deeper exploration of how MCTS specifically enhances the LLM agent's capabilities in a unique way.
- The experimental evaluation, although showing promising results, is somewhat inadequate. The authors do not provide a clear rationale for the selection of the 20 datasets from the AutoML Benchmark (AMLB). Given the diversity of tasks available in AMLB, it would be beneficial to understand the principles behind the choice of these specific tasks to ensure a comprehensive assessment of SELA's capabilities.
- The paper lacks a detailed analysis of the results and the specific contributions of MCTS to the observed performance improvements. A more in-depth discussion on how MCTS influences the exploration and optimization process, compared to other search strategies (beyond random ones), would strengthen the paper's contributions. In particular, UCB typically requires a significant number of exploitation steps to accurately evaluate the value of each node. However, this would incur a huge workload in an AutoML scenario. It would be beneficial to discuss how the authors achieve a good trade-off between efficiency and effectiveness, and why limited exploitation of UCB suffices to gain a performance improvement.

**Questions:**

- The experimental evaluation could be more comprehensive. Further exploration and justification of the chosen datasets, along with a deeper analysis of the results, would enhance the paper's impact and contribution to the field.
- Can you provide details on the size of the search pool for the MCTS procedure? Additionally, how many iterations does MCTS typically require to achieve good performance?

My score may be revised upon reviewing the authors' response to these limitations.

---

> ### Author Response · Authors · 2024-11-22
> **Comment [1/2]**
>
> 1. "While the integration of MCTS with LLM agents is an interesting approach, the innovation appears limited. The application of MCTS in this context is relatively straightforward and does not introduce significant novel techniques beyond existing LLM-agent frameworks like Data Interpreter. The paper could benefit from a deeper exploration of how MCTS specifically enhances the LLM agent's capabilities in a unique way."
>
>     We emphasize that integrating Monte Carlo Tree Search (MCTS) with LLM agents for machine learning tasks is a complex endeavor. MCTS relies on a well-defined search space and depth to enable efficient exploration and exploitation. While several works attempt to integrate MCTS with LLMs, the vastness of the language space often renders critical phases, such as node expansion and simulation, poorly defined, leading to high variance. SELA addresses this by providing a more structured search space tailored to task-specific machine-learning problems, ensuring effective integration with LLM agents.
>
>     Additionally, since simulations require LLMs to generate complete, executable pipelines, and model training can be computationally intensive, achieving strong performance within a limited number of rollouts becomes crucial. To address this, we modify the UCT selection algorithm (line 299-320) to prioritize deeper node exploration and exploitation more rapidly, contributing to the practical adoption of MCTS in compute-intensive scenarios.
>
>     SELA's novelty compared to LLM-based frameworks like Data Interpreter is evident both quantitatively and qualitatively. Table 1 demonstrates SELA's superior performance over Data Interpreter and AIDE. Our experiments reveal that Data Interpreter's feature engineering and model proposals lack sufficient diversity. In contrast, SELA generates a wider range of diverse feature engineering strategies and model training insights, enhancing the overall performance of machine learning tasks.
>
>     For example, while Data Interpreter might provide generic instructions like "Engineer features to improve model performance," leaving the LLM to instinctively generate code, which usually has low diversity and concentrates on a few options, such as polynomial features or PCA. SELA offers detailed and actionable guidance. It proposes specific insights, such as:
>     - Perform time-series analysis on 'PurchDate' to identify any temporal trends in the occurrence of 'Kick' cars and engineer features based on these insights to improve model performance.
>     - Derive new features from the existing ones, such as calculating the click-through rate (CTR) for each ad or user.
>     - Aggregate features to create summary statistics, such as the average credit amount per purpose or the median age per credit history category.
>     - Analyze the distribution of the target variable 'IsBadBuy' to understand the class imbalance, and engineer features that address this imbalance to improve model performance.
>     This level of specificity not only improves the quality of the generated code but also ensures a more diverse and effective approach to feature engineering and model development.
>
> 2. "The experimental evaluation could be more comprehensive. Further exploration and justification of the chosen datasets, along with a deeper analysis of the results, would enhance the paper's impact and contribution to the field."
>
>     We sample 20 datasets that have a number of instances ranging from 500 to 150,000 and a number of features fewer than 500 from AMLB datasets and Kaggle competitions. It is easy for quick experimentation and evaluation, and most of the tasks are also directly associated with industrial applications. Appendix A Table 3 outlines the specific metadata of each dataset used.

---

> ### Author Response · Authors · 2024-11-22
> **Comment [2/2]**
>
> 3. "A detailed analysis of the results and the specific contributions of MCTS to the observed performance improvements."
>
>     **Trade-off Between Efficiency and Effectiveness:**
>
>     We acknowledge the concern regarding the potential workload of UCT in an AutoML scenario. To address this, we modify the UCT algorithm to prioritize deeper nodes earlier in the search process (line 299-320), allowing SELA to explore promising regions of the search space more efficiently. While traditional UCT typically requires extensive exploitation steps to evaluate node values accurately, our adjustments focus on balancing exploration and exploitation within the constraints of a limited number of rollouts. This approach ensures SELA can achieve meaningful improvements without incurring prohibitive computational costs.
>
>     **Search Pool and Iterations:**
>
>     The size of the search pool in SELA depends on the specific problem, as it dynamically adapts to the dataset and task complexity. For our reported experiments, the search pool typically includes hundreds of potential configurations, encompassing diverse strategies for feature engineering, model selection, and optimization. On average, SELA requires around 10 rollouts to achieve significant performance gains, with the option to extend the rollouts for further improvements when computational resources permit.
>
>     **Empirical Results on Extended Rollouts:**
>
>     To demonstrate the relationship between rollouts and performance, we conducted experiments on ablation datasets with extended rollouts. As shown in the table below, increasing the number of rollouts leads to noticeable performance improvements. For instance, with 10 rollouts, the average normalized score (NS) improved from 59.8 to 64.5. These results highlight that while SELA performs well with limited rollouts, additional iterations allow for further refinement and optimization, providing flexibility to balance efficiency and effectiveness based on available resources.
>     By combining these techniques, SELA ensures that limited exploitation of UCT suffices to deliver performance improvements while maintaining computational feasibility in the AutoML setting.
>
>     | Rollout | Avg NS |
>     |---------|---------|
>     | 1       | 59.8    |
>     | 5       | 63.8    |
>     | 10      | 64.5    |
>     | 15      | 65.3    |
>     | 20      | 66.3    |
>
> 4. "A more in-depth discussion on how MCTS influences the exploration and optimization process, compared to other search strategies (beyond random ones), would strengthen the paper's contributions."
>
>     - The inclusion of random search serves to demonstrate that SELA's insights are inherently more effective than the baseline instructions provided by Data Interpreter. As shown in Table 2, even when configurations are selected randomly, SELA outperforms Data Interpreter, highlighting the general utility of SELA's proposed strategies.
>
>     - MCTS is particularly well-suited for navigating the expansive search space required in our setting. Unlike strategies such as BFS, DFS, or A*, which may struggle to handle the requirement for executable pipelines to evaluate scores, MCTS incorporates a simulation phase that facilitates direct score evaluation. This makes MCTS not only feasible but also practical for this context. Additionally, BFS and DFS would involve significantly higher computational complexity, lacking the dynamic balancing of exploration and exploitation that MCTS provides. These factors make MCTS a natural choice for efficiently optimizing in such a complex domain

---

> > ### Comment · Reviewer_epEJ · 2024-11-25
> >
> > I thank the authors for their responses, which partially address my concerns. However, I remain primarily concerned about the novelty and search efficiency of SELA, and therefore will maintain my current score. I will continue to follow the discussion thread.

---

> ### Author Response · Authors · 2024-11-25
>
> ## Novelty
> We appreciate your consideration of SELA’s contributions and the opportunity to clarify its novelty further. While SELA integrates Monte Carlo Tree Search (MCTS), a well-established algorithm, its application and adaptation to AutoML represent a novel advancement. Specifically:
>
> - **Domain-Specific MCTS Adaptation**: SELA customizes MCTS for sequential machine learning pipeline construction, with each tree layer representing distinct ML stages (Data Preprocessing, Feature Engineering, Model Training). Unlike general-purpose MCTS applications, SELA incorporates domain-specific constraints to ensure meaningful exploration of the search space. Moreover, we modify the selection algorithm to ensure nodes at deeper layers can be selected more quickly. This adaptation is not seen in other MCTS applications.
> - **Scalable Solution Generation**: Unlike traditional AutoML frameworks, SELA enables efficient rollout strategies that yield complete, executable ML pipelines at every stage, allowing each exploration step to contribute directly to the final solution.
> - **Empirical Impact**: The experiment results highlight SELA’s ability to achieve competitive results even under limited rollouts, showcasing its efficiency compared to methods requiring extensive search steps.
>
> Together, these aspects distinguish SELA from existing frameworks, demonstrating both its conceptual and practical novelty. We sincerely wish you could provide more specific concerns you might have regarding the novelty of SELA.
>
> ## Search Efficiency
> Thank you for pointing out your concern about the search efficiency. However, as the table in the previous response illustrates, there is a significant performance boost observed within 10 rollouts. This rapid improvement reflects SELA’s ability to prioritize high-reward paths early in the search process through MCTS-guided exploration. Beyond 10 rollouts, the rate of improvement slows due to convergence towards high-performing solutions. This behavior aligns with expectations in constrained search spaces and demonstrates SELA’s efficiency.
>
> We are trying to understand if there is a specific reason you are concerned about the search efficiency of SELA, and we are happy to discuss it further.

---

### Official Review · Reviewer_PPui · 2024-11-02

**Soundness:** 1
**Presentation:** 1
**Contribution:** 1
**Rating:** 1
**Confidence:** 5

**Summary:**

The work introduces an LLM based AutoML tool that uses MCTS to search for better performing configurations.

**Strengths:**

The ablation study showing the strenghts of different base LLMs is interesting.

**Weaknesses:**

The work largely seems to be unfamiliar with AutoML tools and the broader literature. This is particularly evident with the work repeatedly claiming falsehoods about AutoML. For example, early on it is said that AutoML tools typically only focus on the model training aspect, while ignoring feature engineering stages and so on. This claim is easily refuted as many of the cited tools do include data preprocessing and feature preprocessing into account and generally tend to the whole machine learning pipeline. It is often left to the user however what to include in the search space, thus a user might only want to focus on the model training element. Further, the work claims as a contribution that the search is inspired by how humans would conduct hyperparameter search/configuration. I.e., repeated experimentation while refining the search. This is exactly what nearly all AutoML tools do. Many are based on bayesian optimization which typically executes an initial design to probe the search space, then queries the most promising configuration(s) based on an internal model, based on the observation of the actual performance the model is refined and the search continues until a configuration budget is exhausted. Again, this is the standard approach in AutoML, making the claim in "contribution 1" not a contribution of this work. Further, contribution 2 hinges on the use of MCTS for AutoML (with an LLM base). However, MCTS has been already used and studied in the realm of AutoML (see the work by Rakotoarison https://inria.hal.science/hal-01966957/file/MOSAIC-wkpICML2018.pdf) strongly diminishing this claimed contribution. Further, the experiments do not consider an AutoML approach that uses MCTS for comparison.

I was already skeptical towards the experiments due to the untruths written about AutoML. I am further convinced that the evaluation of the aproaches is completely unfair. For AutoGluon and AutoSKLearn it is stated that they are simply used in their default configurations. However, it is not clear how much compute budget each method was allocated and that SELA very likely had a much larger configuration budget available than all other methods. Despite these weird setup issues, AutoGluon shows very strong performance, which is waved aside by the work. An important baseline that needs to be compared against in the AutoML realm is also missing. TabPFN uses a foundation model trained on synthetic data to learn to solve ML tasks and has shown highly competitive or even outperforming classical AutoML systems. This method thus has to be considered in a work as it is presented here. Overall the work can absolutely not be published. The work has to be largely rewritten to remove the false claims and the experiments likely need to be completely redone. I do not see immediate action items that would increase my trust in the work and would make me reconsider my scoring.

**Questions:**

What is the configuration budget allocated to all methods? Since you mentioned AutoSKlearns 600seconds wall-clock time of budget, did you limit SELA to the same small budget?

---

> ### Author Response · Authors · 2024-11-22
> **Comment [1/2]**
>
> We really appreciate your feedback. We respect your rating and your concern regarding our work. Nevertheless, we would still like to address several misunderstandings that seem to underlie your concerns.
>
> # 1. Claims Regarding the Scope of AutoML Contributions
>
> At no point do we state that AutoML tools only focus on model training. Our original wording, “These frameworks primarily focus on optimizing hyperparameters and model ensembling” (lines 36-37), and “Early AutoML efforts… focused primarily on automating key pipeline components like hyperparameter optimization, model selection, and ensembling” (lines 136-138), is a clear acknowledgment of the primary emphasis in these tools.
>
>   **The distinction between data pre-processing and feature engineering:** We also would like to clarify the distinction between data pre-processing and feature engineering based on our setup. Although these two terms are often used together as a means to "construct suitable features from raw input data" [1], we treat them as two separate stages where "data preprocessing" allows the data to be trained by ML models and "feature engineering" encourages better model performance. Although AutoGluon has a module that provides some form of "data pre-processing" before the model training, it only focuses on basic format conversion and missing value handling [2, 3]. We believe that this handling cannot be regarded as feature engineering that helps improve model performance. Moreover, AutoSklearn doesn't have data-preprocessing handling as robust as AutoGluon, let alone feature engineering.
>
>   **Auto-FE:** There are prior works that explore various strategies for automated feature engineering [4, 5, 6]. However, these typically focus on feature space exploration using handcrafted heuristic traversal strategies and generally lack the dynamic adaptivity to task-specific descriptions and datasets that SELA emphasizes. For instance, SELA can propose derived features by leveraging the natural language understanding of dataset columns and task descriptions. While works like CAAFE [1] offer context-aware auto-feature engineering through code generation, it should be noted that CAAFE is designed as a component of the feature engineering process, rather than a comprehensive solution for the entire machine learning pipeline. In contrast, SELA integrates feature engineering seamlessly into a larger, dynamic pipeline that includes model training, improvement, and optimization. This holistic approach allows SELA to adapt to task requirements across all stages of the pipeline, whereas CAAFE focuses primarily on feature engineering without addressing downstream tasks. This distinction underscores SELA’s broader scope and versatility in tackling complex ML workflows.
>
> [1] https://arxiv.org/pdf/2305.03403
>
> [2] https://arxiv.org/pdf/2003.06505
>
> [3] https://auto.gluon.ai/stable/tutorials/tabular/tabular-feature-engineering.html
>
> [4] https://ieeexplore.ieee.org/stamp/stamp.jsp?tp=&arnumber=7344858
>
> [5] https://ieeexplore.ieee.org/stamp/stamp.jsp?tp=&arnumber=7836821
>
> [6] https://link.springer.com/chapter/10.1007/978-3-030-43823-4_10
>
> # 2. Use of Bayesian Optimization in Existing Systems:
>
> Many AutoML tools, such as AutoSklearn, effectively utilize Bayesian optimization and model ensembling for iterative search and refinement, particularly in model selection and hyperparameter tuning. While these methods excel in optimizing model configurations, they often lack the ability to extend adaptive search strategies to other pipeline stages, such as task-aware feature engineering or data preprocessing. In contrast, our approach in SELA dynamically adjusts each stage of the pipeline, providing a more holistic adaptation. Employing Bayesian methods for LLM-based AutoML is certainly a promising direction, but it presents significant challenges, including navigating the expansive natural language search space and parameterizing it for an end-to-end pipeline.

---

> ### Author Response · Authors · 2024-11-22
> **Comment [2/2]**
>
> # 3. MCTS Use and Prior Work:
> We appreciate the mention of Rakotoarison et al.’s MCTS-based AutoML work; we would definitely add this to the related works. However, this method differs fundamentally from SELA’s framework. Their approach utilizes hard-coded parameters and is limited to static configurations, lacking the flexibility needed to dynamically handle task-specific adaptations. In contrast, SELA employs LLMs to construct the search space dynamically, supporting on-the-fly generation of configurations for each pipeline stage and each dataset—something not achieved in prior work. We want to emphasize that although both Rakotoarisan's and our work adopt MCTS, Rakotoarison's essentially operates on a fixed pipeline whilst SELA dynamically generates the whole pipeline code from scratch. Additionally, the results reported in Rakotoarison et al.'s paper show that their approach and AutoSklearn yield very similar performance. However, our experiments suggest that SELA significantly outperforms AutoSklearn, highlighting SELA’s superior performance and adaptability in comparison to both these methods. This difference in performance further emphasizes the value and novelty of SELA’s approach.
>
> # 4. Experiment setup
> **Time Constraint for AutoSklearn and AutoGluon**
> For AutoSklearn, a time constraint of 600 seconds was set because the framework requires specifying a time budget for operation. This choice was made to align closely with the average running time of AutoGluon across all datasets (~10 minutes). By doing so, we aimed to ensure comparability between these two methods under equivalent conditions.
>
> **LLM-Based Method Time Allocation**
> SELA, Data Interpreter, and AIDE are LLM-based methods where a significant portion of the runtime is spent on generating and refining pipeline configurations (potentially more than half of the total time). Consequently, while the traditional AutoML methods focus their time primarily on model training, the LLM-based methods allocate a substantial portion of their runtime to pipeline construction. While we acknowledge that SELA may require more than 600 seconds in some cases due to the non-trivial time taken by LLM API calls, we ensure a strict 600-second execution limit for model training. Notably, despite AIDE not being subject to such a constraint, SELA achieves superior performance, maintaining a win rate of 75% against AIDE across the benchmark tasks. This demonstrates SELA's efficiency and effectiveness even under competitive runtime conditions.
> Nevertheless, to further ensure a comprehensive evaluation, we also conducted experiments with AutoSklearn running under an extended time budget of 3600 seconds. The results indicate that while AutoSklearn's average performance does improve with a longer runtime (with the performance on half of the datasets actually dropping), it still does not match the performance of SELA. This outcome underscores SELA's efficiency and ability to produce high-quality solutions within shorter time frames, despite the inherent challenges of balancing pipeline construction and model training within its runtime.
>
> | Dataset | AutoSklearn 600s | AutoSklearn 3600s | SELA |
> |---------|-----------------|-------------------|------|
> | Boston | 19.5 | 24.2 | 40.1 |
> | Click Prediction Small | 40.2 | 39.9 | 23.2 |
> | colleges | 2.1 | 68.9 | 87.8 |
> | credit-g | 35.1 | 39.0 | 50.9 |
> | GesturePhaseSegmentation | 67.2 | 65.4 | 67.9 |
> | mfeat-factors | 97.1 | 97.0 | 95.7 |
> | **Avg. NS** | 43.5 | 55.7 | 60.9 |
>
> We understand that we may not convince you to change your rating, but we are more than happy and open to additional suggestions to improve our work.

---

> > ### Comment · Reviewer_PPui · 2024-11-25
> > **Response to Rebuttal**
> >
> > Thank you very much for the thorough rebuttal. I acknowledge that I have read all other reviews and rebuttals. I remain unconvinced of the method and keep my score. As I have given the most negative score compared to the other reviewers I am looking forward to the discussion and remain open to their feedback.

---

> > > ### Author Response · Authors · 2024-11-25
> > >
> > > Thank you for your continued engagement and for considering the perspectives of other reviewers. We appreciate your acknowledgment of the rebuttals and your openness to discussion. As your score represents the most critical evaluation, we would greatly value more specific feedback regarding the aspects of SELA that you find unconvincing. In particular:
> > >
> > > - **Novelty:** Are there specific elements of SELA’s design or contributions that you believe lack sufficient differentiation from existing methods? If so, we would welcome suggestions for further clarification or comparisons that could strengthen our case.
> > >
> > > - **Feature Engineering:** In your previous comment, you suggested that we have several false claims regarding the feature engineering of other AutoML frameworks. Does our response convince you that our claims are valid?
> > >
> > > - **Search Efficiency:** While we have elaborated on the performance trends and efficiency observed within the limited rollouts, are there additional metrics, experiments, or details that you feel would better support our claims?
> > >
> > > - **General Feedback:** Are there broader methodological, theoretical, or practical concerns that we may have overlooked? Your insights could help us identify areas for improvement or refinement in future iterations of this work.
> > >
> > > We genuinely value your critical perspective and would greatly appreciate any additional suggestions or explanations you can provide. Your feedback is crucial not only for this submission but also for improving SELA’s contributions to the field.

---

### Official Review · Reviewer_U8rG · 2024-11-04

**Soundness:** 2
**Presentation:** 2
**Contribution:** 3
**Rating:** 5
**Confidence:** 3

**Summary:**

This paper introduces SELA (Tree-Search Enhanced LLM Agents), a novel approach that combines Large Language Model (LLM) agents with Monte Carlo Tree Search (MCTS) for automated machine learning (AutoML). The primary contribution is defining an AutoML pipeline using a tree structure and experimentally demonstrating its effectiveness.

**Strengths:**

The paper presents an interesting pipeline for AutoML based on LLM agents, where the functional attributes of each node layer are intuitively and innovatively defined.

**Weaknesses:**

1. The paper lacks sufficient detail and resources for reproducibility.
2. The study involves multiple modules and details; however, the results in Table 2 appear to show only marginal improvements.
3. Additional issues are mentioned in the "Questions" section.

**Questions:**

1. Regarding the `simulate` phase of MCTS, I understand that your tree has a depth of 6 levels (including the root node). Consequently, $x_{\text{sample}}$ does not necessarily correspond to the final Model Evaluation node, so $c(x_\text{sample})$ does not represent a complete configuration file. How, then, can this be provided to the experimenter $E$ to obtain a score-evaluable code?

2. Regarding lines 205 and 323, when I expand a node at time $\tau_i$, I provide the problem description $p$, dataset information $d$, and $\lambda(x_i)$ (where $\lambda(x_i)$ might concatenate with $\lambda(x_{i-1})$) as the LLM’s prompt, generating $m$ values of $\lambda(x_{i+1})$, from which one is randomly sampled for simulation—is this correct?

3. Given that even the most SOTA models struggle to generate immediately executable code, especially for complex ML tasks, how do you handle cases where the code provided by the LLM fails to execute? Do you assign it a minimum score or apply a different approach?

4. In lines 242–258, could you clarify the distinction between Value and Simulation Score, as well as between Solution Code and Stage Code? My understanding is that Stage Code corresponds to the current node’s code, but is Solution Code a set of codes of child branches? Since the node may have multiple branches.

5. In line 363, "Each framework utilizes the training and validation sets to train models," but I do not see any component in SELA that can be trained.

6. Regarding line 443, as a comparison, how exactly does SELA refine its Solution? I only understand that it finds the optimal solution through MCTS Search, but if a node's result is not good, it seems they won't refine it either, they just won't select it.

7. Line 393 mentions each AutoML rollout is conducted 10 times. Isn’t this count too low for a tree search, potentially resulting in a sparse tree structure?

8. Line 1160 states that each node corresponds to a single feature; however, combining multiple feature engineering techniques is standard practice. Does SELA not support such feature combinations?

---

> ### Author Response · Authors · 2024-11-22
> **Comment [1/2]**
>
> Thank you for your thorough review. We appreciate the detailed questions that will help improve the clarity of our work and would like to take the chance to address each of your comments.
>
> 1. "The paper lacks sufficient detail and resources for reproducibility."
>
>     Please note we list configurations such as dataset split, LLM, temperature, setup for each framework, etc. in 4.1 EXPERIMENTAL SETUP (line 355-397). Important prompts are also provided in Appendix B. SELA is running experiments on a machine equipped with an RTX 3090 (24 GB VRAM), 36-core Intel(R) Core(TM) i9-10980XE CPU @ 3.00GHz, and 125 GB RAM. Each experiment costs approximately $0.05 for LLM API calls (refer to Appendix D, Table 7).
>
> 2. "The study involves multiple modules and details; however, the results in Table 2 appear to show only marginal improvements."
>
>     We respectfully disagree that our method only produces marginal improvements considering the performance difference between AutoML frameworks can be very small across AMLB datasets. Moreover, in head-to-head comparisons, SELA demonstrates substantial superiority by beating all frameworks with a 65%-to-80% win rate. We include a detailed discussion in the general comment (Section 3 Performance compared to the baselines).
>
> 3. "$x_\text{sample}$ does not necessarily correspond to the final Model Evaluation node..." and "how can this be provided to the experimenter E to obtain a score-evaluable code"
>
>     Indeed, $c(x_\text{sample})$ does not necessarily represent the whole pipeline configuration and we ask Experimenter to decide what to do for stages without a specified configuration. For example, $c(x_\text{sample})$ may instruct imputing missing values with mean for data preprocessing and a polynomial expansion for feature engineering, without touching the modeling stage. In this case, Experimenter may pick any model instinctively from LLM's knowledge for the modeling stage and finish the simulation. This implementation is consistent with common MCTS practice, where the simulation is made up of a directional part, represented by the path up to the node chosen, and a random part of any valid moves.
>
> 4. "Regarding lines 205 and 323, when I expand a node at time $\tau_i$, I provide the problem description $p$, dataset information $d$, and $\lambda(x_i)$ (where $\lambda(x_i)$ might concatenate with $\lambda(x_{i−1})$) as the LLM’s prompt, generating $m$ values of $\lambda(x_{i+1})$, from which one is randomly sampled for simulation—is this correct?"
>
>     We assume when you say "time $\tau_i$", you actually refer to a machine learning stage as defined in line 209, for $\tau$ is not a time step in the search process. As described in the Methods section under "Expansion", when a node $x$ at depth $\delta$ is expanded, its child nodes inherit its attributes and receive a new insight $\lambda(x_\text{child})$ specifically for stage $\tau_{\delta+1}$. The insights are pre-generated by the InsightProposer at the beginning of the process (Equation 1), creating the search space $\Lambda$. During expansion, we select from these pre-generated insights rather than generating new ones.
>
> 5. "Given that even the most SOTA models struggle to generate immediately executable code, especially for complex ML tasks, how do you handle cases where the code provided by the LLM fails to execute? Do you assign it a minimum score or apply a different approach?"
>
>     The current SOTA LLM agents are, in fact, capable of generating immediately executable code with a high success rate [1, 2]. Based on these advancements, we leverage LLM agents as the Experimenters for generating and executing code, as noted in 3.2 PIPELINE EXECUTION AND CODE GENERATION. During the simulation, even in the cases where the code fails to execute, the Experimenter can obtain the error message and rewrite the code, further improving the success rate. If the Experimenter fails to generate an executable pipeline within the allowed maximum retries, we will assign a score of 0 to this pipeline configuration, indicating that it may be too challenging to implement and discouraging further exploration of this path.
>
>     [1] https://arxiv.org/pdf/2402.18679
>
>     [2] https://arxiv.org/pdf/2308.08155

---

> ### Author Response · Authors · 2024-11-22
> **Comment [2/2]**
>
> 6. The distinction between Value and Simulation Score, as well as between Solution Code and Stage Code
>
>     - Value ($v(x)$) is initially the simulation score when this node is first simulated, but is updated during backpropagation by averaging with simulation scores from its descendents, following standard MCTS practices
>
>     - Simulation Score ($s(x)$) is the specific performance score from simulating this single node
>
>     - Stage Code ($\sigma_\text{stage}(x)$) contains the code generated up to the current stage ($\tau$). For instance, if the node corresponds to the Feature Engineering stage (denoted as $\tau_3$ as in line 207), then the stage code for this node covers Data Analysis ($\tau_1$), Data Preprocessing ($\tau_2$), and Feature Engineering ($\tau_3$), which is a prefix part of the full solution code of this node. This design renders this convenience: each time a node is about to be simulated, we can reuse the stage code of its parent as the starting point and only generates code afterwards, reducing token usage and enhancing consistency across simulations.
>
>     - Solution Code ($\sigma_\text{sol}(x)$) represents the complete pipeline code after simulation the current node. Please also refer to line 222 and 291 for more explanation.
>
> 7. In line 363, "Each framework utilizes the training and validation sets to train models," but I do not see any component in SELA that can be trained.
>
>     The word "training" refers to training ML models in the generated pipelines. For example, in one simulation, the Experimenter may receive insight such as training a random forest model, generating the corresponding codes, and executing the training process.
>
> 8. "Regarding line 443, as a comparison, how exactly does SELA refine its Solution? I only understand that it finds the optimal solution through MCTS Search, but if a node's result is not good, it seems they won't refine it either, they just won't select it."
>
>     The act of selecting nodes within SELA is indeed both an exploration and a refinement of the pipeline's configuration. The search space in SELA is vast, and we operate under the assumption that it contains configurations capable of achieving high performance. MCTS allows us to iteratively explore this space and refine our search toward such configurations.
>
> 9. "Line 393 mentions each AutoML rollout is conducted 10 times. Isn’t this count too low for a tree search, potentially resulting in a sparse tree structure?"
>
>     We have conducted additional experiments with a larger number of rollouts to evaluate the impact on performance. As detailed in our response to Reviewer epEJ (Q3 Empirical Results on Extended Rollouts), increasing the number of rollouts does indeed lead to improved performance, as the tree structure becomes denser and exploration is more thorough. However, even with 10 rollouts, our method demonstrates substantial performance improvements over the baselines. This result highlights the efficiency of our approach, where even a limited number of rollouts suffices to achieve significant gains.
>
> 10. "Line 1160 states that each node corresponds to a single feature; however, combining multiple feature engineering techniques is standard practice. Does SELA not support such feature combinations?"
>
>     Sorry for not making this clear. While we present the cases as examples, each insight doesn't necessarily mean one feature, method, or approach specifically. In line 200-201, we suggest that "each proposed insight suggests either a single technique or a combination of methods aimed at enhancing performance". In other words, an insight could be multiple strategies of feature engineering. For instance, this is a feature engineering insight proposed for the credit-g dataset: "Derive new features from existing ones, such as calculating the credit-to-income ratio or the age-to-employment duration ratio." This insight suggests that SELA doesn't necessarily create a single feature at each stage, and it is capable of proposing a combination of strategies.

---

> > ### Comment · Reviewer_U8rG · 2024-11-24
> >
> > Thanks for your detailed explanation. I have some interest and concerns regarding the rollout details of MCTS. I would like to confirm: does Rollout = 1 correspond to a greedy result in your response to epEJ? Your tree has a maximum of three layers—Data, Feature, and Model—correct?
> >
> > I think the rollout (selection) time is extremely limited compared to the common MCTS setting, but as you reported, it still works effectively. One important reason could be that your tree structure only has three layers and a limited number of child nodes, right? What is the maximum number of child nodes that can be expanded? Since each rollout expands only one node, the number of leaf/terminal nodes is less than the number of rollouts, correct?
> >
> > Also, since you don’t use a value function, if the expanded child node is not a terminal state, do you use a greedy policy to complete the simulation until the final answer? In your experiments, the results improved steadily from 1 to 20. If you continue scaling, what do you think is the peak result it can achieve? I believe it would be meaningful to explore the scaling boundary, and I look forward to seeing the results.
> >
> > Apologies if you find these technical questions excessive, but I truly believe they are important.

---

> > > ### Author Response · Authors · 2024-11-25
> > >
> > > Thank you for your insightful observations. We appreciate the opportunity to clarify and expand upon these points.
> > >
> > > ## Rollout = 1 and Search Dynamics:
> > > Rollout = 1 simplifies SELA to a non-search-based method, so it is not greedy in nature. Without the MCTS mechanism, Rollout = 1 generates ML pipeline code directly, which is equivalent to Data Interpreter as shown in the table of the general comment (1. Novelty comparing other AutoML frameworks).
> > >
> > > ## Tree Structure and Child Node Constraints:
> > > As detailed in the supplementary materials, we cap the maximum number of child nodes per stage at 5. This results in a search space of $5^{3}$ (125 nodes), given the three layers corresponding to Data Preprocessing, Feature Engineering, and Model Training stages. This limit balances exploration and computational efficiency, while still accommodating a diverse range of potential solutions. The specific value of 5 is a tunable parameter that could vary based on use case or dataset characteristics. We also want to emphasize that the tree search is some kind of "meta-search" in nature, as a single pipeline code can have already involved model ensemble or hyperparameter tuning. The insights provided are much higher level, so the search space doesn't seem as big as the traditional hyperparameter search space. This also explains why the performance increase is significant within limited search steps.
> > >
> > > ## Number of Leaf Nodes and Rollouts:
> > > It is correct that the number of leaf nodes is less than the number of rollouts. However, each node selected during a rollout generates a complete solution code. In other words, a search process of k rollouts produce k candidate solutions with each of them being a complete pipeline. This feature enhances the efficiency and practicality of SELA, as each step delivers a fully executable and evaluable outcome.
> > >
> > > ## Scaling Beyond Rollout = 20:
> > > As shown in the table, the performance improvement slows as the number of rollouts approaches 20. While there is an initial upward trend, the gains diminish due to the convergence of the search process and the increased likelihood of rediscovering similar solutions. Scaling further beyond 20 rollouts may yield marginal benefits while incurring higher computational costs. Exploring optimal scaling thresholds remains an interesting direction for future work.

---

> > > > ### Comment · Reviewer_U8rG · 2024-11-26
> > > >
> > > > Thank you to the authors for the detailed information. I find this work to be an interesting attempt to apply LLM search to specific domains. However, due to a lack of familiarity with related AutoML techniques, and since this aspect raised serious concerns among other reviewers, I cautiously maintain my original score for now. I will closely follow the upcoming discussions and hope the authors continue to improve this interesting work.

---

### Official Review · Reviewer_kFK4 · 2024-11-04

**Soundness:** 3
**Presentation:** 3
**Contribution:** 2
**Rating:** 3
**Confidence:** 3

**Summary:**

The paper proposes a method (SELA) that combines LLMs with MCTS to solve AutoML tasks. More concretely, SELA uses LLMs to generate a search space of solutions (essentially generating different possibilities across different stages in the ML pipeline), which is then explored using MCTS. Through experiments on tasks from the AutoML benchmark and Kaggle competitions, the paper demonstrates SELA's efficacy against baseline methods.

**Strengths:**

- The paper conveys the main ideas and results clearly.
- Handling the NP-hardness of the problem by incorporating LLM agents with a search procedure is intuitive and makes a lot of sense.

**Weaknesses:**

My main concern with this paper is regarding the novelty of the proposed method. The authors note that prior work has already explored combining LLMs with tree search methods. This makes it unclear if the technical contribution of this paper is novel enough for ICLR. The primary contribution of applying LLMs with tree search to AutoML could be an interesting contribution had the paper shown that tree search cannot be trivially combined with LLMs to solve AutoML problems, and highlighted the key challenges faced.

**Questions:**

Could the authors elaborate on some key challenges faced when integrating LLM agents with tree search for AutoML tasks, and how SELA addresses them? All in all, I am trying to understand if the application is straightforward.

---

> ### Author Response · Authors · 2024-11-22
>
> Thank you for the feedback. We appreciate your interest in the contributions of our approach in combining LLMs with tree search for AutoML. We want to emphasize that integrating LLM agents with tree search for AutoML is complex and far from straightforward. Key challenges include:
>
> 1. **Complexity of the Solution Space:** AutoML is inherently NP-hard, with potentially infinite combinations of models, hyperparameters, and feature engineering strategies. Navigating this space efficiently requires advanced methods to identify relevant paths without exhaustive search. SELA addresses this by combining MCTS with LLMs that generate targeted insights based on the task and dataset, making the exploration intelligent and selective.
>
> 2. **Limitations of current LLM-based AutoML frameworks:** The existing LLM-based AutoML methods [1, 2, 3] adopt a single-iteration or greedy refinement approach, which may overlook long-term dependencies or fail to thoroughly explore options. SELA’s iterative, tree-guided process provides a more structured search, allowing the LLM to make decisions that consider both immediate rewards and potential future gains.
>
> 3. **Scalability in Search Strategy:** Due to the large computation requirements, traditional search methods like DFS or BFS are infeasible. MCTS is well-suited for balancing exploration and exploitation, but adapting it to work seamlessly with LLMs requires non-trivial design choices. SELA’s architecture and the selection algorithm are used to ensure that the MCTS is executed within feasible computational bounds, making it possible to achieve high-performance AutoML without exhaustive searches.
> By addressing these challenges, SELA provides a sophisticated integration of LLM agents with MCTS, making the application both innovative and efficient for complex AutoML tasks.
> While tree search has established roots in optimization, integrating it with LLMs for AutoML is non-trivial and, we believe, brings novelty. Adapting Monte Carlo Tree Search (MCTS) to handle the complex solution space and decision dynamics of LLMs in AutoML required overcoming challenges like search depth optimization and computational feasibility. Our work demonstrates how MCTS can work dynamically with LLM agents, achieving an adaptive, efficient framework that standard techniques alone cannot match. We would welcome further guidance on what additional elements might strengthen the perceived novelty of this integration.
>
> [1] https://arxiv.org/pdf/2402.18679
>
> [2] https://www.weco.ai/blog/technical-report
>
> [3] https://arxiv.org/pdf/2304.14979

---

### Author Response · Authors · 2024-11-22
**General Comments [1/2]**

We really appreciate the feedback provided by all reviewers. To further help the audience understand our work, we provide concrete examples of insights and solutions generated by SELA in supplementary materials. In the meantime, we want to address some common issues raised by the reviewers and seek to clarify some points that might be misunderstood.

# 1. Novelty comparing other AutoML frameworks

Given that the experimental results have demonstrated the superior performance of our method against the chosen baselines, we believe it is also beneficial to qualitatively elaborate on the unique values SELA brings about beyond the baselines. We summarize the key differences between SELA and other frameworks in the following table.

|       | Dynamic Pipeline | Adaptive Feature Engineering    | Model Training    | Model Improvement          | Pipeline Optimization    |
|---|---|---|---|---|---|
| AutoGluon | ✗   | ✗    | Fixed models   | Multi-layer stacking + bagging | ✗       |
| AutoSklearn | ✗ | ✗    | Fixed models   | Bayes Opt. + meta-learning + ensemble | ✗   |
| Data Interpreter | ✓   | Instinctive  | Instinctive | Instinctive   | ✗    |
| AIDE  | ✓ | Instinctive | Dynamic & diverse| Dynamic & diverse | One-step refinement + LLM   |
| **SELA**| ✓ |Dynamic & diverse | Dynamic & diverse | Dynamic & diverse |Stepwise MCTS + LLM|

**Table**: Comparison of key capabilities across various AutoML methods. Dynamic indicates the system's ability to adjust workflows based on intermediate outcomes, allowing it to adapt as new information emerges. Diverse refers to employing multiple strategies or methods across tasks, capturing varied modeling needs. Instinctive means that the system directly relies on the decisions generated by an LLM and heavily depends on the model's inclination.

Characterized in the table, LLM Agents like Data Interpreter craft solutions dynamically but are unable to improve them. In contrast, traditional AutoML methods like AutoGluon and AutoSklearn employ optimization strategies but rely on static pipelines. We argue that our paper elegantly overcomes the limitation of both types of methods by mimicking human experts to dynamically construct pipelines and optimize them iteratively. Our contribution may be easier to understand with two detailed comparisons.

**Comparing with existing LLM Agents:**

While SELA leverages Data Interpreter [1] as its Experimenter, their focuses have fundamental differences. Data Interpreter is an LLM agent aiming to produce functional and performant pipelines for ML problems. Its main techniques such as Hierarchical Graph Modeling and Programmable Node Generation are proposed to guarantee successful construction of the pipeline. However, Data Interpreter is unable to further improve the pipeline after the first construction, even given multiple trials. SELA, on the other hand, abstracts away the pipeline-building process assuming a capable Experimenter (section 3.2) and places a crucial emphasis on optimization. The optimization consists of an insight proposer for diverse configurations and a stepwise MCTS process for efficient exploration, allowing continuous discovery of better pipelines, which Data Interpreter falls short of. In fact, when presented with a Data-Interpreter-like agent, it is not obviously clear how we can equip it with an optimization capability. We see limited literature discussing such strategies for ML agents (AIDE [2] counts as one, and we include it in our evaluation comparison and show its performance can be surpassed). By introducing a structural and systematic search process, we provide an intuitive yet highly effective methodology to fill the blank.

**Comparison with traditional AutoML frameworks:**

Our method shares the same important idea of searching with traditional AutoML frameworks, introducing systematic and feedback-driven optimization procedures crucial to acquiring competitive solutions for ML problems. However, our method is distinctive in two ways. First, SELA proposes a dynamic and task-aware search space consisting of natural language insights specifically based on the datasets. For example, when the dataset requires house price prediction, our Insight Proposer may put forward options such as "Create a feature by calculating the age of the house at the time of sale", a semantically meaningful and potentially useful feature operation that we cannot expect from frameworks such as AutoGluon and AutoSklearn. We have included more examples of insights in the supplementary materials for demonstration. Additionally, traditional AutoML frameworks face significant challenges in performing end-to-end ML tasks starting from natural language task descriptions and dataset information to producing final solutions. In contrast, SELA seamlessly executes the entire pipeline: interpreting the task description and dataset, building and optimizing the ML pipelines, and delivering final predictions.

---

> ### Author Response · Authors · 2024-11-22
> **General Comments [2/2]**
>
> # 2. Non-trivial integration with MCTS
> Integrating Monte Carlo Tree Search (MCTS) with LLM agents for ML tasks is far from a straightforward process. MCTS relies on a well-defined and structured search space to effectively balance exploration and exploitation. However, the vastness and flexibility of the language space in LLMs can make key MCTS components, such as node expansion and simulation, ambiguous and prone to high variance.
> SELA addresses these challenges by tailoring the search space to specific machine-learning tasks, ensuring that MCTS operates within a domain where its strengths can be fully leveraged. For example, SELA introduces task-specific constraints and designs structured action spaces that guide node expansion and simulations, enabling more consistent and meaningful outcomes.
> Additionally, the computational demands of simulations in SELA—requiring LLMs to generate complete, executable pipelines followed by model training—necessitate performance optimization within a limited number of rollouts. To achieve this, SELA incorporates modifications to the UCT (Upper Confidence Bounds for Trees) algorithm to prioritize deeper exploration of promising nodes and expedite exploitation, ensuring practicality in compute-intensive scenarios.
> In addition, we include experiments with extended rollouts to demonstrate that SELA could achieve even higher performance if more search steps are allowed (see Comments to Reviewer epEJ Q3: Empirical Results on Extended Rollouts).
>
> # 3. Performance compared to the baselines
> There is a concern suggesting that our method exhibits only marginal improvements compared to other AutoML methods. We respectfully disagree and believe this perception might stem from differing expectations between LLM-based benchmarks and traditional AutoML benchmarks.
>
> In AutoML evaluations, as shown in the AMLB paper [3] (Tables 4–15), the performance differences between competing frameworks are often small—sometimes just one or two percentage points or even within a single percentage point. Such differences are typical, given that AutoML frameworks often operate in highly optimized and competitive settings.
>
> While it is not unexpected for AutoML frameworks to show similar average normalized scores (NS), SELA’s superior average best ranking and the high win rate is a strong indicator of its ability to construct better ML pipelines and achieve superior results across a diverse range of datasets. This ranking metric reflects SELA's effectiveness not just in achieving high performance but in doing so consistently across various tasks.
>
> [1] https://arxiv.org/pdf/2402.18679
>
> [2] https://www.weco.ai/blog/technical-report
>
> [3] https://arxiv.org/pdf/2207.12560

---

### Meta-Review · Area_Chair_z5gD · 2024-12-08

**Metareview:**

This paper proposes to do a specific variant of MCTS for AutoML tasks, where the possible actions consist of standard data-science actions, e.g. data processing, feature engineering, proposing models, etc.

Comparisons are made against standard AutoML packages such as AutoGluon, AutoSklearn, with generally competitive results.

## Strengths:
* The main message of the paper, from my perspective, is showing that LLMs can perform agentic approaches for AutoML problems, which has the potential for future work in e.g. data science, scientific discoveries, AI for AI, etc.
* If we assume that the experiments were rigorously done, then results seem comparable to SOTA AutoML platforms (AutoGluon, etc.)

## Weaknesses:
* There was an overall sentiment of "It's just another LLM can do XYZ" paper.
* In my opinion, the writing style is pretty opaque - Section 3 I think has a poor combination of notation and wordiness, that made it difficult to read.

Unfortunately, none of the reviewers wishes to champion this paper, as the highest score as a 5. The overall message "LLM Agents can do AutoML" wasn't well-received, as most of the negative feedback came from novelty and experimental rigorousness.

In general, this paper might not be suitable for ICLR, and I actually recommend the authors send this paper to a LLM-focused conference, provided that they rewrite the paper to focus more on the overall agentic message. Otherwise, I can forsee it having the same feedback issues if submitted to NeurIPS/ICML (lack of novelty, etc.). This is unless, somehow the proposed method would lead to significantly better experimental outcomes.

**Additional Comments On Reviewer Discussion:**

There are a few points raised, which aligns with what I mentioned above, where this paper may be better suited for more specific LLM conferences (e.g. COLM 2025).

The main points raised were:
* Possibly unfair / unrigorous comparisons against baselines, from a AutoML-specific perspective (Reviewer PPui)
* Reproducibility / lack of specific details on the search and how it affects performance (Reviewers U8rG, epEJ)
* Lack of novelty - i.e. "It's just MCTS" (Reviewers kFK4, epEJ)

These points are completely understandable, and the authors did attempt to address all of these points in expected ways (e.g. explaining what is novel, claiming experimentation was done fairly, etc.), but this didn't change any of the reviewers' minds.

---

### Decision · Program_Chairs · 2025-01-22

Reject